# LAION-5B: An open large-scale dataset for training next generation image-text models

**Christoph Schuhmann**[1] §§°°     **Romain Beaumont**[1] §§°°     **Richard Vencu**[1,3,8] §§°°
**Cade Gordon**[2] §§°°     **Ross Wightman**[1]§§     **Mehdi Cherti** [1,10]§§
**Theo Coombes**[1]     **Aarush Katta**[1]     **Clayton Mullis**[1]     **Mitchell Wortsman**[6]
**Patrick Schramowski**[1,4,5]     **Srivatsa Kundurthy**[1]     **Katherine Crowson**[1,8,9]
**Ludwig Schmidt**[6] °°     **Robert Kaczmarczyk**[1,7] °°     **Jenia Jitsev**[1,10] °°
LAION[1]     UC Berkeley[2]     Gentec Data[3]     TU Darmstadt[4]     Hessian.AI[5]
University of Washington, Seattle[6]     Technical University of Munich[7]     Stability AI[8]
EleutherAI[9]     Juelich Supercomputing Center (JSC), Research Center Juelich (FZJ)[10]
`contact@laion.ai`
§§ Equal first contributions, °° Equal senior contributions

## Abstract

Groundbreaking language-vision architectures like CLIP and DALL-E proved the utility of training on large amounts of noisy image-text data, without relying on expensive accurate labels used in standard vision unimodal supervised learning. The resulting models showed capabilities of strong text-guided image generation and transfer to downstream tasks, while performing remarkably at zero-shot classification with noteworthy out-of-distribution robustness. Since then, large-scale language-vision models like ALIGN, BASIC, GLIDE, Flamingo and Imagen made further improvements. Studying the training and capabilities of such models requires datasets containing billions of image-text pairs. Until now, no datasets of this size have been made openly available for the broader research community. To address this problem and democratize research on large-scale multi-modal models, we present LAION-5B - a dataset consisting of 5.85 billion CLIP-filtered image-text pairs, of which 2.32B contain English language. We show successful replication and fine-tuning of foundational models like CLIP, GLIDE and Stable Diffusion using the dataset, and discuss further experiments enabled with an openly available dataset of this scale. Additionally we provide several nearest neighbor indices, an improved web-interface for dataset exploration and subset generation, and detection scores for watermark, NSFW, and toxic content detection. [1]

## 1   Introduction

Learning from multimodal data such as text, images, and audio is a longstanding research challenge in machine learning [31, 51, 56, 83, 86]. Recently, contrastive loss functions combined with large neural networks have led to breakthroughs in the generalization capabilities of vision and language models [58, 59, 66]. For instance, OpenAI's CLIP models [58] achieved large gains in zero-shot classification on ImageNet [65], improving from the prior top-1 accuracy of 11.5% [41] to 76.2%. In addition, CLIP achieved unprecedented performance gains on multiple challenging distribution shifts [3, 23, 61, 70, 78, 82]. Inspired by CLIP's performance, numerous groups have further improved image-text models by increasing the amount of computation and the training set size [28, 54, 89, 94]. Another recent success of multimodal learning is in image generation, where DALL-E [59] and later

---

[1]Project page: https://laion.ai/laion-5b-a-new-era-of-open-large-scale-multi-modal-datasets/

36th Conference on Neural Information Processing Systems (NeurIPS 2022) Track on Datasets and Benchmarks.

models [52, 60, 64, 66, 90] demonstrated the potential of text-guided image generation by producing high-quality images specific to the provided text.

A critical ingredient in this new generation of image-text models is the pre-training dataset. All of the aforementioned advances rely on large datasets containing hundreds of millions or even billions of image-text pairs, e.g., 400 million for CLIP [58] and 6.6 billion for BASIC [54]. However, *none of these datasets are publicly available*. While OpenAI still released the CLIP models publicly [58], later papers made neither the pre-training dataset nor the resulting models available to the wider research community [2, 28, 52, 54, 66, 89, 90]. As a result, research in this area has pooled into a small number of industrial research labs, limiting transparency and impeding research progress.

In this work, we address this challenge and make multimodal training more accessible by assembling a public dataset that is suitable for training large image-text models. Specifically, we introduce LAION-5B, the largest public image-text dataset containing over 5.8 billion examples (see Table 1 for a comparison). By starting from Common Crawl [1] and filtering this data source with an existing CLIP model, we derive a dataset consisting of three parts: 2.32 billion English image-text examples, 2.26 billion multilingual examples, and 1.27 billion examples that are not specific to a particular language (e.g., places, products, etc.). Beyond assembling the dataset, we also explore its ethical implications and flaws that emerge with large-scale data collection. By releasing LAION-5B publicly, we offer the first opportunity for the community to audit and refine a dataset of this magnitude.

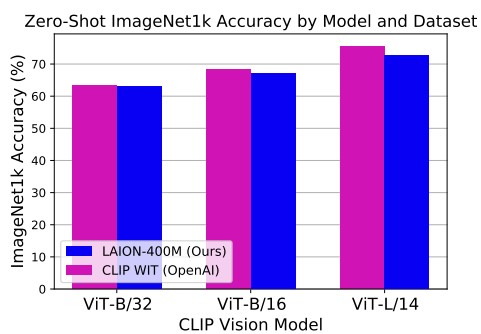

| Dataset | # English Img-Txt Pairs |
|---|---|
| **Public Datasets** | |
| MS-COCO | 330K |
| CC3M | 3M |
| Visual Genome | 5.4M |
| WIT | 5.5M |
| CC12M | 12M |
| RedCaps | 12M |
| YFCC100M | 100M[2] |
| **LAION-5B (Ours)** | **2.3B** |
| **Private Datasets** | |
| CLIP WIT (OpenAI) | 400M |
| ALIGN | 1.8B |
| BASIC | 6.6B |

Figure 1: **Zero-Shot Accuracy.** CLIP models trained on LAION-400M (ours) [69], a previously released subset of LAION-5B, show competitive zero-shot accuracy compared to CLIP models trained on OpenAI's original training set WIT when evaluated on ImageNet1k.

Table 1: **Dataset Size.** LAION-5B is more than 20 times larger than other public English image-text datasets. We extend the analysis from Desai et al. [14] and compare the sizes of public and private image-text datasets.

To validate that LAION-5B is indeed suitable for training large image-text models, we conduct multiple experiments. We focus on matching the performance of OpenAI's CLIP models because they are the largest publicly released image-text models. OpenAI's CLIP models were trained on 400 million image-text pairs, and hence we also train CLIP models on a subset of LAION-5B containing the same number of examples ("LAION-400M"). Across a diverse range of problem settings including ImageNet (zero-shot), distribution shifts, VTAB, retrieval, and fine-tuning, our models trained on LAION-400M match or come close to the performance of OpenAI's CLIP models. Our ViT-L/14 models trained with OpenCLIP are the first open source reproductions of the largest CLIP models released by OpenAI.

Despite these validation results, LAION-5B is *not* a finished data product. Due to the immense size of current image-text pre-training datasets, curating LAION-5B for widespread use goes beyond the scope of a single research paper. Hence we do not only release our dataset, but also our software stack we built for assembling LAION-5B. We view our initial data release and this paper as a first step on the way towards a widely applicable pre-training dataset for multimodal models. As a result,

---

[2]Although YFCC100M contains 100M image-text pairs, it is unclear how well the text matches the image for an average example from the dataset. Radford et al. [57]'s curation procedure reduced YFCC100M to 15M samples.

**we strongly recommend that LAION-5B should only be used for academic research purposes in its current form. We advise against any applications in deployed systems without carefully investigating behavior and possible biases of models trained on LAION-5B.**

The remainder of the paper proceeds as follows. After reviewing related work, we present our data collection process for LAION-5B in Section 3. Section 4 then describes LAION-5B's composition including its various subsets. To validate LAION-5B, we reproduce and evaluate different image-text models in Section 5. Before concluding, we discuss the technical limitations of LAION-5B in Section 6 and safety and ethics concerns in Section 7.

## 2    Related Work

**Vision-Language Models.** Radford et al. [58] made a large step forward in multimodal learning for image-text data with their CLIP (Contrastive Language–Image Pre-training) model. The authors proposed a contrastive learning scheme to embed both images and text into a shared representation space, which enabled unparalleled performance in zero-shot image classification. Moreover, CLIP made large progress on multiple challenging distribution shifts [78, 84].

After CLIP's initial success, ALIGN and BASIC improved contrastive multimodal learning by increasing the training set size and the batch size used for training [28, 54]. LiT also increased training scale and experimented with a combination of pre-trained image representations and contrastive fine-tuning to connect frozen image representations to text [94]. Flamingo introduced the first large vision-language model with in-context learning [2]. Other papers have combined contrastive losses with image captioning to further improve performance [43, 89]. Beyond image classification and retrieval, the community later adapted CLIP to further vision tasks such as object navigation and visual question answering [17, 32, 50, 72].

Another direction that has recently seen large progress in multimodal learning is text-guided image generation [47, 62, 95]. Specifically, DALL-E demonstrated diverse image generation capabilities for text prompts combining multiple concepts [59]. GLIDE, DALL-E 2, Imagen, Parti, and Stable Diffusion then improved visual fidelity and text-prompt correspondence [52, 60, 64, 66, 90].

**Image-Text Datasets.** Earlier dataset creation efforts such as MS-COCO and Visual Genome curated image and region labels through human annotation [36, 44]. While this resulted in high-quality labels, it also limited the scale of the datasets to only 330K and 5M examples, respectively. The web-harvested YFCC-100M dataset is substantially larger with about 99 million images and one million videos from Flickr, but only contains the user-generated metadata without additional annotations collected specifically for training computer vision models [79]. As a result, the text associated with an image sometimes has little to no correspondence with the actual image content.

To address this shortcoming of web-harvested image-text data, the Conceptual Captions dataset (CC3M) started with images and alt-text collected from the web, but then performed additional data cleaning procedures [71]. To increase the size of the dataset, researchers later relaxed the filtering protocol to arrive at the subsequent CC12M dataset [11]. Building datasets from alt-text continued with ALT200M [26] and ALIGN [28], which increased the dataset size up to 1.8 billion image-text pairs. In contrast to relying on alt-text, RedCaps used the captions provided by Reddit users to collect higher quality captions [14].

Datasets with non-English image-text pairs are less common. As a result, researchers translated English captioning datasets to other languages such as Farsi, Korean, and Japanese [67, 73, 74]. To the best of our knowledge, the largest multilingual dataset before LAION-5B has around 36 million samples from Wikipedia Image Text [75]. With the release of LAION-5B, researchers now have access to roughly two orders of magnitude more multilingual samples, which provides new opportunities for research on low-resource languages and multilingual models.

**Scaling Behavior.** Improving model performance by increasing data scale has been a theme in machine learning since at least the ImageNet dataset [13]. In the following decade, computer vision benefited from growth in model, data, and compute scale, in addition to advances in both convolutional and transformer architectures [15, 33, 81, 92]. Industrial research labs assembled large internal datasets such as Instagram-1B, JFT300M, and JFT3B to support image pre-training [46, 77, 93]. Natural language processing (NLP) demonstrated the beneficial effect of model, data, and compute scale on generalization through large language models such as GPT-3 [8] and associated

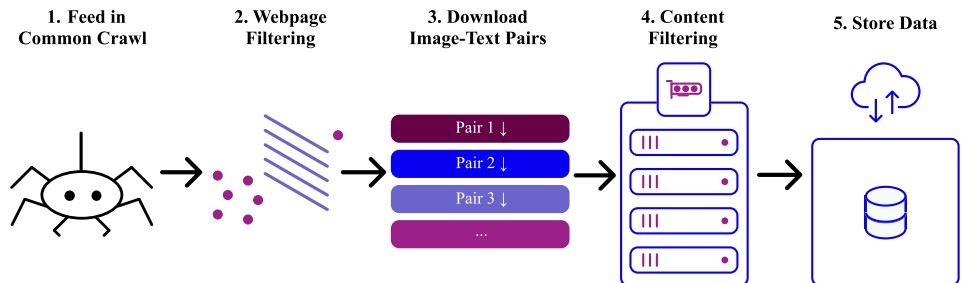

Figure 2: **Overview of the acquisition pipeline:** Files are downloaded, tracked, and undergo distributed inference to determine inclusion. Those above the specified CLIP threshold are saved.

experiments on scaling behavior [30]. Community efforts like the The Pile [18] and BigScience ROOTS [40] made large text datasets more accessible.

## 3    Collection Methodology

We constructed LAION-5B starting from Common Crawl, a public web archive [1]. The Common Crawl organization crawls the web since 2008 and publishes the results in snapshots approximately every month. Recent snapshots each contain about 300 TiB of data for around 3 billion web pages. In the following, we introduce our pipeline to assemble and filter a vision-language dataset from images in Common Crawl and their associated HTML alt-text.

### 3.1    Dataset Assembly Pipeline

Our dataset assembly pipeline follows the flowchart of Figure 2. At a high level, the pipeline consists of three main components: (i) distributed filtering of the Common Crawl web pages, (ii) distributed downloading of image-text pairs, and (iii) content filtering. The code used for the dataset pipeline may be found on GitHub[3]. We now describe each component in more detail.

**Web page filtering.** To extract image-text pairs from Common Crawl, we parse the HTML `IMG` (image) tags from Common Crawl's WAT metadata files.[4] Specifically, we focus on images with an *alt-text* so we can create image-text pairs. The alt-text is an HTML attribute of `IMG` tags containing alternative text for situations where the corresponding image cannot be rendered. For instance, screen reader software for a visually impaired person may read the alt-text in place of an image, or a search engine may use the alt-text to better index a web page without analyzing the actual image content.

After extracting the alt-text, we perform language detection using CLD3 [53] with three possible outputs: English, another language, or no detected language (i.e., all detections are below a confidence threshold [69]). Based on a manual inspection of a random sample, the "no language" set contains language-agnostic short form text such as the names of products and places.

We stored the resulting data in a PostgreSQL server for processing in the next stages of the pipeline. We maintained about 500M image URLs in the server at all times.

**Downloading Image-Text Pairs.** In order to maximize resource utilization, we downloaded the raw images from the parsed URLs with asynchronous requests using the Trio and Asks Python libraries. To limit costs, we chose a small cloud node with 2 vCPUs, 1GB of RAM, and 10Mbps download bandwidth as a worker instance. Such a worker can process 10,000 links in about $10 - 15$ minutes. We utilized roughly 300 workers in parallel and batched the workload into chunks of 10,000 links taken from the aforementioned PostgreSQL server.

**Post-Processing.** After downloading the WAT files from Common Crawl, we removed data with less than 5 characters of text, less than 5 KB of image data, and potentially malicious, large, or redundant images. To conclude the pipeline, we filtered image-text pairs based on their content. Specifically,

---

[3]`https://github.com/rvencu/crawlingathome-gpu-hcloud`
[4]See `https://commoncrawl.org/the-data/get-started/` for details of the metadata format.

Q: An armchair that looks like an apple

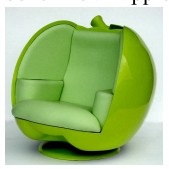

C: Green Apple Chair

Q: A dog rolling in the snow at sunset

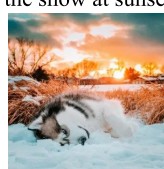

C: sun snow dog

Q: A graphic design color palette

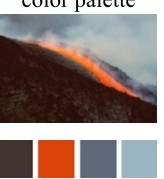

C: Color Palettes

Q: pink photo of Tokyo

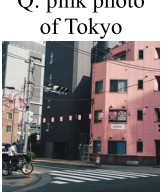

C: pink, japan, aesthetic image

Figure 3: **LAION-5B examples.** Sample images from a nearest neighbor search in LAION-5B using CLIP embeddings. The image and caption (C) are the first results for the query (Q).

we computed cosine similarities between the image and text encodings with OpenAI's ViT-B/32 CLIP model. For languages other than English, we utilized the multi-lingual CLIP ViT-B/32 from Carlsson et al. [10]. While OpenAI also released larger CLIP models later, these models were not available when we began to assemble LAION-5B. For consistency, we therefore relied on ViT-B/32 CLIP models for the entire dataset. We removed all English image-text pairs with cosine similarity below 0.28, and all other pairs with similarity below 0.26. This step removed around 90% of the original 50 billion images, leaving just short of 6 billion examples.

## 3.2 Safety During Collection

Current automated filtering techniques are far from perfect: harmful images are likely to pass, and others are likely to be falsely removed. We make a best effort to identify, document, and tag such content. In the case of illegal content, we computed CLIP embeddings to filter out such samples. Furthermore, these images and texts could amplify the social bias of machine learning models, especially ones trained with no or weak supervision [76]. It is important to note that the above mentioned classifiers are not perfect, especially keeping the complexity of these tasks and the diverse opinions of different cultures in mind. Therefore, we advocate using these tags responsibly, not relying on them to create a truly safe, "production-ready" subset after removing all potentially problematic samples. For a detailed discussion in this regard, we refer to Sec. 7.

To encourage research in fields such as dataset curation, we refrain from removing potentially offensive samples and tag them instead. The user can decide whether to include content depending on their task. To this end, we also encourage model developers to state, e.g., in their model card [49] which subsets and tagged images are used.

We apply Q16 [68] and our own specialized pornographic and sexualized content classifier (here referred to as NSFW) to identify and document a broad range of inappropriate concepts displaying not only persons but also objects, symbols, and text, see *cf.* [68] and Appendix Sec. C.5 and Sec. C.6 for details. Both classifiers are based on CLIP embeddings. Following our main intention of a publicly available dataset, these two approaches, as with all other implementations related to LAION 5B, are open-sourced.

We separate pornographic content and otherwise inappropriate content (e.g. harm, exploitation and degradation). Both can be dis- and enabled in the publicly available dataset exploration UI.[5] With both together, the UI and the openly accessible code, we encourage users to explore and, subsequently, report further not yet detected content and thus contribute to the improvement of our and other existing approaches.

## 4 Dataset Composition

We release LAION-5B as the following three subsets:

- 2.32 billion English image-text pairs. We refer to this subset as LAION-2B-en or LAION-2B if the language is clear from context.

---

[5]https://knn5.laion.ai/

- 2.26 billion image-text pairs from over 100 other languages. In the multilingual subset, the top-5 most frequent languages are Russian (10.6%), French (7.4%), German (6.6%), Spanish (6.6%), and Chinese (6.3%).

- 1.27 billion samples where a language could not be clearly detected. Based on visually inspecting a random subset of these low-confidence language samples, the corresponding images often depict products or places. The captions contain language with clear semantics, but might also include noise such as keywords for search engine optimiziation or product tags.

We provide metadata files in the Apache Parquet format that consist of the following attributes for each image-text pair:

- A 64-bit integer identifier
- The URL of the image.
- The text string.
- Height and width of the image.
- Cosine similarity between the text and image embeddings.
- The output from our NSFW and watermark detectors (one score between 0 and 1 each).

3% of images were detected as NSFW, which can be filtered out by a user with the NSFW tag.

## 5 Experiments Validating LAION-5B

In this section, we showcase prior work using the LAION-400M [69] and other subsets as well as our CLIP reproduction studies to give quantitative and qualitative evidence of the dataset's utility for training SOTA large scale language-vision models.

### 5.1 Usage Examples

**Subdataset Generation.** LAION-5B's scale enables novel dataset curation for computer vision related tasks. Recently, researchers have utilized both LAION-5B and a subset, LAION-400M, as a data source in vision related tasks such as facial representation learning [96] and invasive species mitigation [38]. Within LAION, we have compiled from LAION-5B both LAION-High-Resolution[6], a 170M subset for superresolution models, and LAION-Aesthetic[7], a 120M subset of aesthetic images, as determined by a linear estimator on top of CLIP.

**CLIP Reproduction and Improvements.** Gao et al. [19], trained an enhanced CLIP architecture on the LAION-400M subset, outperforming OpenAI's CLIP on ImageNet zero-shot classification top-1 accuracy. See Sec. 5.2 for our CLIP reproduction experiments using models of different scales. Training on a LAION-5B subset, Li et al. [42] developed BLIP to unify understanding and generation for vision-language tasks via a novel Vision-Language Pretraining (VLP) framework. It has been shown that BLIP matched or outperformed comparable models as per CIDEr, SPICE, and BLEU@4 metrics. Eichenberg et al. [16] used a LAION subset for MAGMA, a model generating text "answers" for image-question pairs; MAGMA achieves state of the art results on OKVQA metrics and outperforming *Frozen* [80].

**Image Generation.** Rombach et al. [63] utilized a subset of LAION-5B in training latent diffusion models (LDM) that achieved state-of-the-art results on image inpainting and class-conditional image synthesis. The work was further extended into stable diffusion project that used subsets of LAION-5B (LAION-2B-en, laion-high-resolution and laion-aesthetics[8]) for training a publicly available SOTA text-to-image generative model (see Appendix Sec. F.2). Furthermore, Gu et al. [21] used LAION-400M to train VQ diffusion text-to-image generation models, which have been shown to be more efficient, and are able to generate higher quality images. Moreover, Saharia et al. [66] showed an improved architecture of a diffusion model that was trained on a subset of LAION-400M that outperforms OpenAI's recent DALLE-2 and achieves a new state-of-the-art COCO FID of 7.27.

---

[6]https://huggingface.co/datasets/laion/laion-high-resolution
[7]https://github.com/LAION-AI/laion-datasets/blob/main/laion-aesthetic.md
[8]See `https://github.com/CompVis/stable-diffusion` for more details

## 5.2 Experiments on CLIP Reproduction

In an effort to reproduce the results of CLIP [58], and to validate the data collection pipeline we describe in Sec. 3, we trained several models on LAION-400M [69] and a model on LAION-2B-en, datasets which are both subsets of LAION-5B. As training such models require large compute due to dataset and model sizes that are considered in the experiments, the usage of supercomputers and large compute clusters is necessary in order to train the models efficiently.

We used OpenCLIP [27], an open source software for training CLIP-like models. After adapting OpenCLIP for distributed training and execution on JUWELS Booster supercomputer [29], we reproduced CLIP models of different size on the LAION-400M subset. We trained ViT-B/32, ViT-B/16, and ViT-L/14 following CLIP [58], and an additional model that we call ViT-B/16+, a slightly larger version of ViT-B/16. We followed the same hyper-parameter choices of the original CLIP models. We used between 128 and 400 NVIDIA A100 GPUs to train the models. All trained models may be found in the OpenCLIP repository[9]. For more information about hyper-parameters and training details, see Appendix Sec. E.1.

### 5.2.1 Zero-Shot Classification and Robustness Performance

Following CLIP [58] and subsequent works, we evaluate the models on zero-shot classification. For each downstream dataset, we use a set of pre-defined prompts for each class, which we collected from prior works [58, 94]. We compute the embeddings of each class by averaging over the embedding of the prompts, computed each using the text encoder. For each image, and for each class, we compute the cosine similarity between their embeddings, and classify each image as the class that have the largest cosine similarity with the image embedding. We evaluate the models using top-1 accuracy.

In Tab. 2, we show a comparison between models trained on LAION (400M, 2B) and original CLIP from [58]. We follow [94] and evaluate robustness performance on ImageNet distribution shift datasets [3, 23, 25, 61, 82]. Additionally, we construct a benchmark we call VTAB+, a superset of VTAB [91], on which we compute the average top-1 accuracy over 35 tasks[10]. We can see that on ImageNet-1k (noted "INet" on the table), performance of LAION-400M models and original CLIP models (trained on a 400M private dataset) is matched well. On the four ImageNet distribution shift datasets, we observe some larger differences, notably on ObjNet (CLIP WIT is better) and INet-S (LAION is better), which allows us to conclude that in overall, CLIP models trained on LAION match in their robustness original CLIP. With ViT-B/32 and ViT-L/14, training on the larger LAION-2B-en improves over LAION-400M model everywhere . Overall, on VTAB+, performance of LAION and CLIP WIT models are similar, except on ViT-L/14, where we observe an advantage of CLIP WIT.

To obtain an idea about how the zero-shot performance improves with scale, we show the relationship between the total compute and accuracy on VTAB+ on models trained on LAION (400M, 2B-en). In Figure 4, we see that accuracy on VTAB+ improves with compute (log-log plot). It would be interesting to study in future work if the relationship between compute and accuracy keeps showing the same trend or whether we start to see saturation, like it was observed in [93]. Here, we can report that increasing either model or data scale for CLIP pre-training results in improvement of zero-shot classification performance on various downstream transfer targets. For a full overview of zero-shot classification and retrieval results, view Sec. E.3 of the Appendix.

To show that larger dataset scale matters for the performance of pre-trained models, we perform additional experiments using ViT-B/32 and ViT-L/14 on different LAION-5B and LAION-400M subsets, while varying the amount of training compute (samples seen). Our findings confirm that the effect of dataset scale is significant, given sufficient compute for training. For instance, for the same amount of compute (34B images seen), training ViT-L/14 on LAION-2B-en (75.4%) outperforms LAION-400M (73.9%) on ImageNet-1k zero-shot classification. Same effect is observed for smaller ViT-B/32 model. For more detailed results, see Fig. 12 and Tab. 6 in the Appendix.

---

[9]https://github.com/mlfoundations/open_clip

[10][91] showed that different aggregation strategies have high rank correlation (Kendall score) with the simple top-1 average accuracy over datasets, thus we follow the same strategy. We also compute the ranks of each model on each task and average the ranks, and find that the ranking is similar to averaging top-1 accuracy.

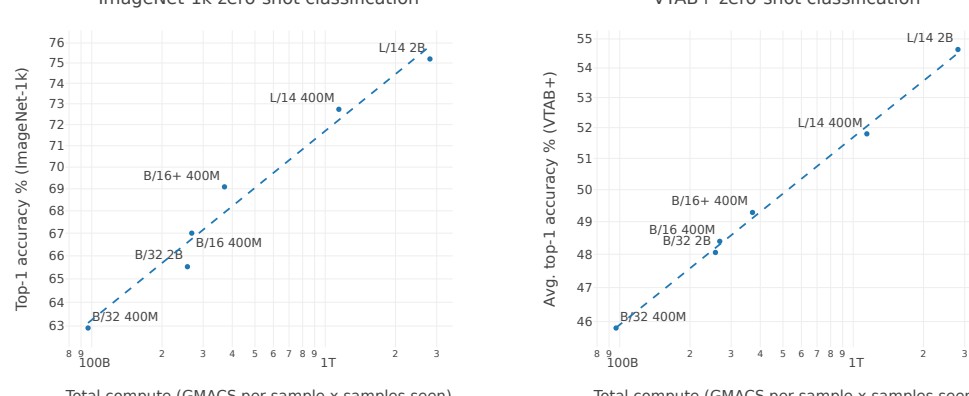

Figure 4: The relationship between total compute (giga multiply–accumulates (GMACS)) and zero-shot top-1 classification accuracy (%) of models trained on LAION (400M, 2B-en). The dashed line in each figure is a linear fit in log-log space. Each point corresponds to a model trained on either the 400M or 2B-en LAION subsets. We show results on ImageNet-1k (left) and VTAB+ (right) where we average the accuracy over 35 tasks (see Appendix E.3 for details). Clear effect of model, data and compute training scale is evident on zero-shot performance that increases following scale power law.

| Model | Pre-training | INet | INet-v2 | INet-R | INet-S | ObjNet | VTAB+ |
|-------|-------------|------|---------|--------|--------|--------|-------|
| B/32 | CLIP WIT | 63.3 | 56.0 | 69.4 | 42.3 | 44.2 | 45.4 |
| | LAION-400M | 62.9$^{-0.4}$ | 55.1$^{-0.9}$ | 73.4$^{+4.0}$ | 49.4$^{+7.1}$ | 43.9$^{-0.3}$ | 45.6$^{+0.2}$ |
| | LAION-2B-en | 65.7$^{+2.4}$ | 57.4$^{+1.4}$ | 75.9$^{+6.5}$ | 52.9$^{+10.6}$ | 48.7$^{+4.5}$ | 47.9$^{+2.5}$ |
| B/16 | CLIP WIT | 68.3 | 61.9 | 77.7 | 48.2 | 55.3 | 47.5 |
| | LAION-400M | 67.0$^{-1.3}$ | 59.6$^{-2.3}$ | 77.9$^{+0.2}$ | 52.4$^{+4.2}$ | 51.5$^{-3.8}$ | 48.3$^{+0.8}$ |
| B/16+ | LAION-400M | 69.2 | 61.5 | 80.5 | 54.4 | 53.9 | 49.2 |
| L/14 | CLIP WIT | 75.6 | 69.8 | 87.9 | 59.6 | 69.0 | 55.7 |
| | LAION-400M | 72.8$^{-2.8}$ | 65.4$^{-4.4}$ | 84.7$^{-3.2}$ | 59.6 | 59.9$^{-9.1}$ | 51.8$^{-3.9}$ |
| | LAION-2B-en | 75.2$^{-0.3}$ | 67.7$^{-2.0}$ | 87.4$^{-0.5}$ | 63.3$^{+3.7}$ | 65.5$^{-3.6}$ | 54.6$^{-1.2}$ |

Table 2: Comparison between CLIP models trained on LAION (400M, 2B) and the original CLIP models [58] trained on OpenAI's WebImageText (WIT) dataset. We show zero-shot top-1 classification accuracy (%) on various datasets including ImageNet, four ImageNet distribution shift datasets, and a benchmark we call VTAB+, where we average performance over 35 tasks. See Appendix E.3 for more details about the datasets used for evaluation and the results.

## 5.3 Experiments with Generative Models

To validate LAION-5B as a dataset for training strong text-to-image generation models, we fine-tuned OpenAI's GLIDE [52] on LAION-5B data. The obtained results comparing generated samples from original OpenAI GLIDE and from our reproduction (LAIONIDE) are compiled into an interactive web demo[11]. See Appendix Sec F for more technical details on experiments with GLIDE (F.1) and Stable Diffusion (F.2).

---

[11]https://wandb.ai/afiaka87/glide_compare/reports/laionide-v3-benchmark--VmlldzoxNTg3MTkz

# 6 Technical Limitations

The large scale of current image-text datasets makes it infeasible to thoroughly investigate all aspects of a dataset in a single publication. Hence we now outline some potential technical limitations specifically affecting LAION-5B. These potential limitations are starting points for future work on analyzing and improving image-text datasets.

**Data Overlap.** Our experiments in Section 5.2 show that models trained on LAION-5B achieve good performance on a variety of downstream tasks. However, the LAION-5B training set may overlap with some of the downstream test sets if these test sets are also included in Common Crawl. If overlap is present, it may lead to incorrectly large test set accuracies that overstate the true generalization capabilities of models trained on LAION-5B.

Overall, we do not consider potential test set overlap to be a serious threat for the validity of results obtained with LAION-5B. OpenAI encountered the same question in the context of their pre-training dataset for CLIP and found only few examples of substantial performance difference due to data overlap on downstream target datasets [58]. Some datasets such as ObjectNet [3] are likely not contained in Common Crawl because ObjectNet was not assembled from web images. Instead, the authors of ObjectNet tasked MTurk workers to take new pictures in their own homes. Nevertheless, measuring the degree of overlap between LAION-5B and popular computer vision benchmarks is an important question for future work, which will include further de-duplication efforts.

**Other text sources.** Birhane et al. [6] described the shortcomings of alt-text and noted that alt-text is not necessarily a good description of the corresponding image. For instance, the alt-text may be search engine optimization (SEO) spam, an incoherent list of keywords, or overly corrupted otherwise. In such cases, the language in the text annotations may become less informative or entirely useless for training. For ImageNet zero-shot classification, BASIC [54] has demonstrated strong results when turning 5 billion of the 6.6 billion captions into the form of `CLASS_1 and CLASS_2 and ... and CLASS_K`, by using an internal multi-label classification dataset (JFT-3B). Thus, image captions formed by just concatenating class names may also serve as meaningful alternative of otherwise corrupted text. Such a finding adds a possibility of employing generated together with existing natural language captions for training contrastive image-language models with strong zero-shot performance.

**Filtering with CLIP.** CLIP allows the curation and collection of this dataset to be low-cost and scalable. Such an automated process reduces dramatically necessity for the human control which would be otherwise intractable for such large scale collection. However, through curating with CLIP, we also incur its flaws and model biases. For additional discussion of CLIP filtering related to safety and ethics, see Appendix Sec. G.2.

Filtering by a small scale CLIP ViT-B/32 may leave more image-text pairs with weak or no semantic connection in the dataset while also accidentally removing some high quality image-text pairs than filtering with stronger, larger scale models that were not available in the time of our experiments. The larger CLIP ViT-L/14 model may create a less noisy version of LAION datasets than what was possible with smaller scale CLIP ViT-B/32. We hypothesize that filtering Common Crawl with a CLIP ViT-L model will further increase the quality of our dataset. It is subject to our future work to create a CLIP ViT L/14 filtered version of LAION-400M and LAION-5B to test how this affects model training and downstream transfer performance.

# 7 Safety and Ethical Discussion

Recent developments in large-scale models, such as GPT-3 [9], CLIP [57], ALIGN [28], GLIDE [52], and DALLE-2 [60] have potential for far-reaching impact on society, both positive and negative, when deployed in applications such as image classification and generation, recommendation systems, or search engines. Besides model parameter scaling, the advances made so far also rely on the underlying large-scale datasets. Recent research [4, 5] described many potential negative societal implications that may arise due to careless use of vision-language models, e.g., the models perform worse for certain groups of users or reproduce discriminatory behavior.

Unfortunately, only a minority of these models are publicly released, most of them are only accessible by an "input to output" interface. Importantly, the underlying large-scale datasets are also not often publicly available. While open-source efforts exist to re-implement model architectures and

training, the closed nature of large-scale datasets used for model training makes any proper systematic investigation of model training and model behavior very hard or even impossible. Studying full training, comparison of different model architectures and progress in large-scale multi-modal learning becomes restricted to those institutions that were able to obtain their closed large-scale datasets. It also results in safety issues of creating and using such models, as broad research community does not get to test both model and the dataset used for its training for causes underlying undesired behaviours.

LAION-5B as an open large-scale dataset provides here not only a chance to make progress in careful studies of the trained models' capabilities and replication but also to investigate how uncurated large-scale datasets impact various model biases and under which circumstances their usage may result in undesired safety issues. Such research can help to design automated ways to curate and create datasets from uncurated ones that alleviate the bias and safety issues. To this end, LAION also created a number of tools to aid researchers and other users in large-scale data handling and exploration. One such a tool uses pre-computed image embeddings to enable search of images guided either by text or image input via an easily and publically accessible web interface (CLIP retrieval tool[12], see Appendix Sec. C.4). LAION made also source code for the tool and routines necessary to build an own version of it publicly available[13] (see Appendix Sec C, C.2, C.3 for more details).

After the release of LAION-400M, several groups (e.g., [6]) already used such tools and investigated potential problems arising from an unfiltered dataset. Motivated by these findings, with LAION-5B, we introduced an improved inappropriate content tagging (*cf.* Sec. 3.2) as well as a watermark filter, which can improve the safety and quality of the text-to-image models trained on the dataset.

Such development indicates that this dataset acts as a starting point, and is not the final endpoint, for creating further improved datasets to train models for various tasks. In our opinion, this process is not supposed to be a non-transparent closed-door avenue. It should be approached by broad research community, resulting in open and transparent datasets and procedures for model training. Towards meeting this challenge, the large-scale public image-text dataset of over 5.8 billion pairs and further annotations introduced here provides diversity that can be a starting point for ensuring balance and for selecting safe, curated subsets for corresponding target applications. We encourage everybody to participate in this exciting and important future journey.

In the current form, we consider this dataset a research artefact and strongly advocate **academic use-only** and advise careful investigation of downstream model biases (Appendix Sec. G.2). Additionally, we encourage users to use the described tools and to transparently explore and, subsequently, report further not yet detected content and model behaviour to our dataset repository[14], and help to further advance existing approaches for data curation using the real-world large dataset introduced here.

**Privacy.** We comment on privacy issues arising from Common Crawl as source of links in LAION-5B and measures undertaken to handle those in the Appendix Sec. G.1

# 8 Conclusion

By releasing LAION-5B, a larger updated version of an openly available dataset that contains over 5 billion image-text pairs, we have further pushed the scale of open datasets for training and studying state-of-the-art language-vision models. This scale gives strong increases to zero-shot transfer and robustness.

To validate the utility of LAION-5B, we demonstrated that a subset of our dataset can be used to train SOTA CLIP models of various scale that match the strong zero-shot and robustness performance of the original models trained on closed curated data, or to fine-tune generative models like GLIDE, producing samples of good quality. The dataset thus provides opportunities in multi-language large-scale training and research of language-vision models, that were previously restricted to those having access to proprietary large datasets, to the broader research community. Finally, thanks to its large scale, even a rather strict subset filtering (driven by various criterion like NSFW, watermark presence, resolution) provides high-quality datasets that are still large enough to provide sufficient scale for the training or fine-tuning of strong specialized language-vision models.

---

[12]https://knn5.laion.ai
[13]https://github.com/rom1504/clip-retrieval
[14]https://github.com/laion-ai/laion5b-bias

## Acknowledgments

We thank Phil Wang, the creator of the DALLE-pytorch github repository[15], who inspired us and helped creating our open community. We also want to thank Aran Komatsuzaki, Andreas Köpf, Bokai Yu, John David Pressman, Natalie Parde, Gabriel Ilharco, Fredde Frallan (see also Appendix) and all the members of the LAION discord server[16] for helping crawling image-text-pairs and run inference on their private computers. We want to thank Hugging Face and Stability AI for their continuous financial support and providing hosting space for open datasets and models. We would also like to thank openAI for making their pre-trained CLIP models publicly available, which allowed us to filter the LAION datasets . We would like to express gratitude to all the people who are working on making code, models and data publicly available, advancing community based research and making research more reproducible.

The authors gratefully acknowledge the Gauss Centre for Supercomputing e.V. [17] for funding this work by providing computing time through the John von Neumann Institute for Computing (NIC) on the GCS Supercomputer JUWELS Booster [29] at Jülich Supercomputing Centre (JSC). We also acknowledge storage resources on JUST [20] granted and operated by JSC. Patrick Schramowski acknowledges the support by the Hessian Ministry of Higher Education, Research, Science and the Arts (HMWK) cluster project "The Third Wave of AI".

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
