# OpenReview forum: "LAION-5B: An open large-scale dataset for training next generation image-text models"
_NeurIPS.cc/2022/Track/Datasets_and_Benchmarks — NeurIPS 2022 Datasets and Benchmarks _

### Official Review · Reviewer_hzjq · 2022-07-25
**A good paper releasing 5.8B image-text pairs dataset potentially very impactful for research on multimodal models**

**Rating:** 9
**Confidence:** 3

**Strengths:**

- The authors have built and made available to the research community the largest dataset of image and text pairs. Moreover this dataset is not limited to 2.32 billion pairs of image and english descriptions but also contains 2 other subsets : 2.26 billion multilingual descriptions, and 1.27 billion salient but non-concrete language samples.
- The release of such an artifact is particularly important for the research community. As the authors indicated in their paper, many recent scientific advances have demonstrated the usefulness of datasets of this nature of several hundred thousand image and text pairs. Nevertheless in these previous research publications these datasets have not been released and the information about their content is limited. This dataset, whose constitution is technically complicated and costly, will allow more research groups to work on 1) models requiring text and image pairs as input and 2) audit and refinement of this dataset.
- The very large size of the dataset allows future searches to look for the best possible cleaning filters or filters.
- A series of experiments showing that the content of the dataset does indeed allow access to similar performances of CLIP [34].
- The distribution of all the codes used to constitute such a dataset. This represents huge engineering work.
- The addition of metadata for each example of the dataset (LANGUAGE, NSFW, similarity, watermark, ...).

**Weaknesses:**

 I very much appreciate the presence of section 5: "Experiments Validating the Utility of LAION-5B". Nevertheless, after reading Figure 4 and Table 2, three questions come to my mind:
1. why not have carried out the training of B/16, B/16+ and L/14 on LAION-2B-en?
2. what would have been the performance of B/32, B/16, B/16+ and L/14 if the training with LAION-400m had been continued during several epochs in order to reach the same amount of compute as the training on LAION-2B-en?
3. Why the Lit method has been used for training CLIP models on Laoin? It makes the comparison of their performance with the original CLIP model less impactful than if the training method had been identical.

**Additional Feedback:**

Thanks to the authors for creating and sharing these artifacts which I think will be very useful for future research.

I have distilled several comments, suggestions for improvement and my questions in the dedicated sections above.

You indicated that "We release the first open-source reproduction of CLIP models up to ViT L/14 scale achieving competitive zero-shot classification and retrieval", I may have missed something but I didn't find the points of this trained model, do you have/plan to open source it?

**Clarity:**

The paper is overall well written.

However, there are a few points that I think could use some clarification:
-  Figure 2 could in my opinion be improved. As it stands, the reader may find it difficult to associate a text with the corresponding arrow/symbol. In this figure there is a mention of "Bloom filters" which are not mentioned (under this term) in the rest of the paper and the step of calculating the CLIP scores is not clearly indicated (even if one could guess that it is the "GPU Inference" step). One way to improve the reading of this scheme might be to make the temporal reading of the pipeline from left to right or from top to bottom.
- LAION-400M is mentioned both as a reference [42] and as a subset of LAION-5B. Indeed, [42] is mentioned in the section "3. Collection Methodology" as a source of inspiration to fix some pipeline filters but also used in the section "5. Experiments Validating the Utility of LAION-5B", notably in section 5.2.1 where the authors state that "CLIP models trained on LAION match in their robustness original CLIP". In order to fully understand the results shown in Table 2, I think the reader would benefit a lot from knowing the relationship between LAION-400M and LAION-5b: is it a uniform random subset? If not, what are the specific criteria that distinguish this subset?

**Correctness:**

The authors' claims seem to me to be correct. The limitations that could be seen with this method - notably the use of the CLIP score and the biases that could thus be reproduced or the use of data from Common Crawl - are well discussed by the authors.

**Documentation:**

The documentation is globally satisfactory,

It just seems to me that two point could be clarified :
- In the part concerning the data collection, the authors indicate "To create image-text pairs, we parse the HTML IMG tags containing an alternative description from Common Crawl's WAT files". For Laoin-5b, were all the Common-crawl dumps used or were only some of the dumps parsed - if so, which ones? Moreover, since one of the advantages of LAOIN-5b is its very large size allowing future searches to apply their own filters, did you keep in the metadata the original dump from which each image and text pair came? A useful case I can see for this type of metadata is to be able to deduplicate pairs of identical images that would have remained identical between 2 versions of a web page (as the same web page can be archived at several dates in Common Crawl).
- In the abstract, they say "Additionally we provide several nearest neighbor indices, an improved web-interface for exploration and subset generation, and detection scores for watermark, NSFW, and toxic content detection", I'm guessing these are accessible in different ways on the "https://huggingface.co/datasets/laion" page. I may have missed something but a page explaining where each of these items can be found would be beneficial to the documentation of this dataset from my perspective.

**Ethics:**

The authors have discussed this topic well in sections 3.2 and 7. Nevertheless, as the choice was to not remove from the dataset examples that could be "potentially offensive samples" in order to "encourage research in fields such as dataset curation" - which for me makes sense - and knowing the pairs came from Common Crawl dumps, I think that this work justifies further ethical discussion or review.

**Relation To Prior Work:**

- The authors have well discussed the relationship of their work with the previous contributions in section 2.
- As mentioned in the review section "Clarity", the relation of this paper with reference [42] would certainly benefit from being clarified.
- In section 2, under "Image-Text Datasets", it seems to me that the set of datasets mentioned is not made available to the scientific community. It would seem to me that this should be specified in this section to justify the impact of the contribution made by LAION-5b.

**Summary And Contributions:**

- Creation and distribution of a dataset composed of 5.8B pairs of images and texts – in several languages - in order to propose to the research community an artifact that has not been shared by scientific publications that have shown the usefulness of such a dataset.
- Sharing of different features related to this dataset facilitating its use: web interface allowing to search in it, sharing of several nearest neighbor indices, detection scores for watermark, NSFW, and toxic content detection
- Experiments showing that the constituted dataset allows to reach similar performances as the ones reported by [34]

---

> ### Author Response · Authors · 2022-08-22
> **Review response**
>
> We thank the reviewer for their positive feedback and recognizing the importance of our datasets. We now respond to each of the points raised by the reviewer:
>
> > why not have carried out the training of B/16, B/16+ and L/14 on LAION-2B-en?
>
> We agree that these are important experiments and are planning to pursue them in future work. While our compute budget for this submission was substantial with 320k GPU hours, it was unfortunately not sufficient for training more models at the LAION-2B scale. Specifically, a single L/14 training run on LAION-2B would have already consumed 250k GPU hours, leaving no time for exploring hyperparameter choices etc.  Hence we instead decided to explore the LAION-400m scale in more depth with several models and train a single B/32 model on LAION-2B.
>
> > what would have been the performance of B/32, B/16, B/16+ and L/14 if the training with LAION-400m had been continued during several epochs in order to reach the same amount of compute as the training on LAION-2B-en?
>
> We thank the reviewer for this interesting question. As mentioned in the previous point, compute constraints unfortunately made it impossible to comprehensively investigate what happens when we train on LAION-400m with the same compute budget as LAION-2B training. We view this as an important direction for future work for which we hope to secure new compute resources (see also response to same question by reviewer XLgX).
>
> As a first step in this direction, we have carried out an additional experiment where we train CLIP models on different subsets of LAION-400m. In particular, we trained an openCLIP ViT B/32 model on a subset of size 80 million images drawn from LAION-400m, where we used the same total compute budget as for training on 400 million images (160 epochs for 80m to match 32 epochs used for LAION-400m). The downstream zero-shot accuracy of the resulting model is still substantially below training with 400 million images in this case (54.89% for B/32 on 80m (160 epochs) vs 62.9% on LAION-400m (32 epochs). Here we provide the figure showing experiments conducted with various fractions of LAION-400m data, including the 80m experiment with extended 160 epochs budget:
>
> [Fractions of LAION-400m Experiments Figure](https://media.discordapp.net/attachments/947179319267065977/986213724660592680/data_vs_accuracy.png)
>
> This provides evidence that data scale matters - even with increased compute the model trained with smaller 80m scale is bottlenecked by the data scale and stays clearly below the model trained on 5x larger 400m scale, compute training being equal.
>
> Currently we are in the process of running the experiment the reviewer suggested with a B/32 model, training it on LAION-400m for 80 epochs (which corresponds to 16 epochs used for B/32 training on LAION-2B). We hope to be able to report the results in the final manuscript, but due to the scale of the experiment we unfortunately cannot promise this.
>
> > Why the Lit method has been used for training CLIP models on Laoin? It makes the comparison of their performance with the original CLIP model less impactful than if the training method had been identical.
>
> We apologize for any confusion. While OpenCLIP does support freezing (locking) the image tower, we did not use the LiT method for training CLIP models on LAION. We agree that keeping the training method identical is important for the performance comparison, which is why we did not incorporate training modifications proposed after the original CLIP paper in our training runs. We will clarify this in our manuscript.
>
> > Figure 2 could in my opinion be improved. As it stands, the reader may find it difficult to associate a text with the corresponding arrow/symbol. In this figure there is a mention of "Bloom filters" which are not mentioned (under this term) in the rest of the paper and the step of calculating the CLIP scores is not clearly indicated (even if one could guess that it is the "GPU Inference" step). One way to improve the reading of this scheme might be to make the temporal reading of the pipeline from left to right or from top to bottom.
>
> We agree that the figure does not describe the collection method in a clear manner. To address this, we have taken your advice to simplify the figure and use left-to-right as a temporal axis for clarity:
> [Updated Figure](https://drive.google.com/file/d/1QPWVQv0xKXb18A_0_HH-Cn_ALy5-SGz-/view?usp=sharing)

---

> > ### Author Response · Authors · 2022-08-22
> > **Review response, Part 2**
> >
> > > LAION-400M is mentioned both as a reference [42] and as a subset of LAION-5B. Indeed, [42] is mentioned in the section "3. Collection Methodology" as a source of inspiration to fix some pipeline filters but also used in the section "5. Experiments Validating the Utility of LAION-5B", notably in section 5.2.1 where the authors state that "CLIP models trained on LAION match in their robustness original CLIP". In order to fully understand the results shown in Table 2, I think the reader would benefit a lot from knowing the relationship between LAION-400M and LAION-5b: is it a uniform random subset? If not, what are the specific criteria that distinguish this subset?
> >
> > LAION-400M is fully contained in LAION-5B, but is not a uniformly random subset. When scaling up our dataset collection from 400M to 5B, we adjusted some aspects of our pipeline. For instance, we changed the CLIP filtering threshold from 0.3 to 0.28. There were also additional small changes in our deduplication code and in the time range of images we accessed. We believe that these changes have little impact on the data distribution and will verify this either for the updated version of our submission or in future work.
> >
> > > As mentioned in the review section "Clarity", the relation of this paper with reference [42] would certainly benefit from being clarified.
> >
> > The LAION-400m reference [42] was an initial release manuscript describing the existence of this dataset and did not come with any validation experiments demonstrating that it is actually possible to train large image-text models on LAION. It is worth noting that this was not a given since the LAION dataset curation process differs substantially from the original dataset creation process OpenAI employed for the CLIP pre-training dataset.
> >
> > Since LAION is a community-driven effort, we decided to publish such an initial manuscript in order to make the machine learning community aware of this effort and invite researchers to join and build high-quality datasets that enable CLIP training. Our current submission provides systematic validation experiments of LAION, the larger LAION-5B dataset, additional analyses of the datasets, and more details on the dataset curation process. This is reflected in the lengths of the respective manuscripts: [42] contains five pages while our submission (with supplementary material) contains 32 pages.
> >
> > We will clarify the relationship between [42] in our submission in more detail in the updated version of our manuscript.
> >
> > > In section 2, under "Image-Text Datasets", it seems to me that the set of datasets mentioned is not made available to the scientific community. It would seem to me that this should be specified in this section to justify the impact of the contribution made by LAION-5b.
> >
> > In the updated version of our manuscript, we will explicitly state the public or closed nature of available large-scale datasets. As a preview, please find a screenshot of our updated table here: https://drive.google.com/file/d/1uegkFvIbwu-2XW_leD_LSPwVcQQm7WDF/view?usp=sharing
> >
> >
> > > In the part concerning the data collection, the authors indicate "To create image-text pairs, we parse the HTML IMG tags containing an alternative description from Common Crawl's WAT files". For Laoin-5b, were all the Common-crawl dumps used or were only some of the dumps parsed - if so, which ones? Moreover, since one of the advantages of LAOIN-5b is its very large size allowing future searches to apply their own filters, did you keep in the metadata the original dump from which each image and text pair came? A useful case I can see for this type of metadata is to be able to deduplicate pairs of identical images that would have remained identical between 2 versions of a web page (as the same web page can be archived at several dates in Common Crawl).
> >
> > We used all available dumps at the time up to October 2021. While we did not keep all metadata, we already followed the deduplication scheme the reviewer suggested. We utilize the image URLs to deduplicate images so that the final dataset contains no re-crawled images.

---

> > > ### Author Response · Authors · 2022-08-22
> > > **Review response, Part 3**
> > >
> > > > In the abstract, they say "Additionally we provide several nearest neighbor indices, an improved web-interface for exploration and subset generation, and detection scores for watermark, NSFW, and toxic content detection", I'm guessing these are accessible in different ways on the "https://huggingface.co/datasets/laion" page. I may have missed something but a page explaining where each of these items can be found would be beneficial to the documentation of this dataset from my perspective.
> > >
> > > Thank you for pointing this out, we will improve the documentation in the updated version of our dataset. The relevant dataset page on HuggingFace is https://huggingface.co/datasets/laion/laion2B-en-joined . This dataset contains the column “punsafe”, which is the output of our NSFW predictor https://github.com/LAION-AI/LAION-SAFETY
> > >
> > > Further detail can also be found in our blog post: https://laion.ai/blog/laion-5b/
> > >
> > > > You indicated that "We release the first open-source reproduction of CLIP models up to ViT L/14 scale achieving competitive zero-shot classification and retrieval", I may have missed something but I didn't find the points of this trained model, do you have/plan to open source it?
> > >
> > > We thank the reviewer for pointing out a missing reference to our publicly available open-source models. Our models are available in the OpenCLIP repository: https://github.com/mlfoundations/open_clip  We will clarify this in the paper.

---

> > > > ### Comment · Reviewer_hzjq · 2022-08-26
> > > > **Thanks for addressing the comments**
> > > >
> > > > Thank you for taking my comments into consideration and addressing them all individually. My concerns have been addressed.
> > > >
> > > > As a result of your response and the elements you plan to change, I am convinced that the quality of the paper will increase significantly. I will increase my grade accordingly.
> > > >
> > > > I would like to note that I suspected that some experiments were not performed due to lack of computational resources but as this was not specified in the paper it remained an assumption. I give a lot of value to the additional (expensive!) experiments you have done/are doing. I find that their conclusion gives an additional argument for the usefulness of the dataset you propose.
> > > >
> > > > Congratulations for this work!

---

### Official Review · Reviewer_Wvxh · 2022-07-26
**Open-source image-text pair dataset of unprecented scale**

**Rating:** 9
**Confidence:** 4
**Clarity:** Yes

**Strengths:**

Open-sourcing a dataset of this scale will enable many researchers to work on vision-text problems. This is especially important since the latest SOTA models in many vision-text fields are all models pretrained on huge amounts of image-text or video-text pairs. In this context, another strength is to not only open-source the dataset but also the collection process and trained models.

The creation of the VTAB+ benchmark is not discussed in detail in the paper, but as a superset of VTAB it can be assumed that VTAB+ is a better estimate of zero-shot classification performance in various tasks than VTAB.

A single model was pretrained on LAION-2B-en and shows the usefulness of the dataset over LAION-400M. This is an achievement given the scale of model, datasets and compute required (400 A100 GPUs for 88 hours). Downstream experiments and their analysis are detailed.

The ethical implications are well discussed and the authors explain their choices on releasing a dataset containing potentially harmful content in a convincing manner. The benefits of the work outweigh the risks.

**Weaknesses:**

In table 2, the original CLIP VIT model is compared with the reproduction on LAION. The authors state in the introduction "numerous groups have further increased CLIP’s generalization ability". My question would be why there is no comparison to these improved CLIP models in this table 2, but only a comparison to the original model. The reasons for choosing only this model and no others should be stated explicitly, so that the reader can assert that the comparison and claims are fair and valid.

In chapter 5.2, all models except one are trained on LAION-400M and only one model is trained on the new LAION-2B-en. The reasons on why the models are not trained on LAION-2B-en (e.g. too high compute cost) should be stated explicitly. In table 2, especially the experiment of model L/14 pretrained on LAION-2B-en is missing, as there the original CLIP WIT model significantly outperforms the CLIP LAION-400M model and the best performance of the original L/14 CLIP WIT is never reached.

The new VTAB+ benchmark is mentioned briefly but lacking details. What are the reasons for creating it, why is it more useful than VTAB? How are the text prompts exactly created, how does the downstream evaluation pipeline look exactly? Unless I missed it, the benchmark and evaluation code is not public yet. At least the supplemental should contain these details in a separate chapter as well as an outlook on when/how the benchmark and evaluation code will be made public, since the "open-sourced-ness" of this work is its main strength.

**Additional Feedback:**

Typo on page 6: "Additionnaly"

**Correctness:**

The experimental claims are correct and can be reproduced with the OpenCLIP code.

The dataset construction process is sound and also documented in detail.

The experiments are appropriate and correct. As stated in the weaknesses, the claims on the VTAB+ benchmark are currently hard to understand and verify due to missing details and code but seem believable and sound.

**Documentation:**

Yes

**Relation To Prior Work:**

Yes

**Summary And Contributions:**

The main contribution is a new large-scale open-source 5.85 billion image-text pair dataset named LAION-5B. The dataset is a folloup to LAION-400M. Both datasets were crawled from the web and filtered with CLIP with the goal to keep only pairs where images and text are related to each other.

The authors describe the collection process and properties of the resulting dataset. Next, they explain and show usage examples for LAION-5B and LAION-400M, both new contributions by the authors (subdataset generation) and contributions from other works (CLIP reproduction, generative vision-language tasks).

The main experiments in this work are reproductions of various CLIP models on the LAION-400M and LAION-2B-en datasets and their comparison to the original CLIP model trained on the private WebImageText dataset (CLIP WIT).

The authors evaluate the models on zero-shot top-1 classification accuracy (%) on ImageNet and 4 datasets with distribution shifts compared to ImageNet (e.g. ObjectNet was "collected to intentionally show objects from new viewpoints on new backgrounds", quote from https://objectnet.dev/ ).

Additionally, the authors scale up to VTAB benchmark with 19 tasks to VTAB+ with 35 tasks and evaluate on this benchmark. VTAB and VTAB+ contain natural tasks (classical vision like CIFAR), specialized tasks (remote sensing like areal images or medical images) and structured tasks which require counting objects or predicting their orientation.

Zero-shot classification evaluation is done using predefined text prompts for each class and selecting the class where the prompt has the highest predicted similarity to the image.

Experimental results show: 1) It is possible to match the performance of the original CLIP VIT model when training on LAION datasets. 2) The model trained on 2B images improves over 400M images. 3) Zero-shot performance of models does not seem to saturate yet when increasing compute, so further upscaling may improve results further.

Finally there is an extensive discussion on technical limitations, safety, ethics and biases. The authors choose to flag images that are predicted to be NSFW instead of removing them and remove only illegal images.

---

> ### Author Response · Authors · 2022-08-22
> **Review response**
>
> We thank the reviewer for the very positive review. Below, we respond to each of the weaknesses mentioned by the reviewer.
>
> > In table 2, the original CLIP VIT model is compared with the reproduction on LAION. The authors state in the introduction "numerous groups have further increased CLIP’s generalization ability". My question would be why there is no comparison to these improved CLIP models in this table 2, but only a comparison to the original model. The reasons for choosing only this model and no others should be stated explicitly, so that the reader can assert that the comparison and claims are fair and valid.
>
> We would like to clarify that the purpose of Table 2 is to provide a direct comparison of our LAION-trained CLIP models to OpenAI’s CLIP models where we match as many training aspects as possible (optimizer, architecture, etc.) - except the training set. This allows us to measure the effect of changing the training set without confounding from the other training aspects. The fact that our accuracy numbers are close to those of the OpenAI CLIP models validates the usefulness of LAION-5B for training image-text models.
>
> We agree that there have been efforts after the original CLIP paper that further improved the performance achievable by image-text training, e.g., BASIC and CoCa. However, these models use different architectures, larger training sets, or other training algorithms. Hence, comparing these models does not allow us to isolate the effect of the training dataset.
>
> Nevertheless, we appreciate the reviewer’s suggestion and will extend Table 2 in the final version of our paper to include the current state-of-the-art models for each test set. This will give the reader a more comprehensive picture of how our models compare to other reported accuracy numbers.
>
> > In chapter 5.2, all models except one are trained on LAION-400M and only one model is trained on the new LAION-2B-en. The reasons on why the models are not trained on LAION-2B-en (e.g. too high compute cost) should be stated explicitly. In table 2, especially the experiment of model L/14 pretrained on LAION-2B-en is missing, as there the original CLIP WIT model significantly outperforms the CLIP LAION-400M model and the best performance of the original L/14 CLIP WIT is never reached.
>
> We agree with the reviewer and will state in the text explicitly that the main reason for missing LAION-2B experiments with larger model scales are compute limitations. We exhausted our available compute budget after training openCLIP L/14 on LAION-400m and could not afford to train a ViT L/14 model on 2B images. Currently we attempt to obtain further compute resources to train an openCLIP L/14 model on LAION-2B. We will provide the results if the experiments finish before the end of the discussion period, but cannot promise that this will be the case since obtaining compute resources at this scale is non-trivial.
>
>
> > The new VTAB+ benchmark is mentioned briefly but lacking details. What are the reasons for creating it, why is it more useful than VTAB? How are the text prompts exactly created, how does the downstream evaluation pipeline look exactly? Unless I missed it, the benchmark and evaluation code is not public yet. At least the supplemental should contain these details in a separate chapter as well as an outlook on when/how the benchmark and evaluation code will be made public, since the "open-sourced-ness" of this work is its main strength.
>
> We thank the reviewer for making us aware of the incomplete information. We will make the composition of VTAB+ more explicit in the text. The dataset composition is provided in Table 5 of the supplementary material. We will also highlight that VTAB+ is assembled from existing VTAB and ImageNet robustness datasets.
>
> The benchmark description and code are already publicly available at  https://github.com/LAION-AI/CLIP_benchmark, with notebook demonstrating evaluation at https://github.com/LAION-AI/CLIP_benchmark/blob/main/benchmark/results.ipynb We will update the manuscript to contain corresponding pointers. The exact prompts used for zero-shot transfer evaluation and further details be seen in this python file: https://github.com/LAION-AI/CLIP_benchmark/blob/182d106cd04c1f36f264a25e5c41796c01ae7748/clip_benchmark/datasets/builder.py#L383

---

> > ### Comment · Reviewer_Wvxh · 2022-08-26
> > **Thanks**
> >
> > Thank you for your response, I am satisfied by the response and will keep my review at the current score of 9.

---

### Official Review · Reviewer_XLgX · 2022-07-27
**A strong contribution to the vision-language community which could benefit from additional analysis**

**Rating:** 8
**Confidence:** 3
**Clarity:** The paper is well written.

**Strengths:**

The two primary strengths of this dataset are its size and its openness. With regard to size, there are no other datasets of similar (or even close to similar) size that are available to the community. Collecting a dataset of this scale requires a significant engineering effort.

As emphasized throughout the paper, all aspects of the data downloading, processing, and tagging pipeline are open-source. These could be used to reproduce/update the dataset and to collect specialized datasets in the future. In general, the openness of the project is a refreshing change from other large-scale vision-language accessible only by API.

This paper also makes a few minor contributions which should not go unmentioned:
 - The img2dataset tool is already widely used
 - The watermark detection model is useful to the community
 - The VTAB+ dataset is a slightly larger version of VTAB

**Weaknesses:**

The primary weakness of the paper is that there should be a more in-depth analysis of the dataset contents and filtering methodology.

It is not reasonable to expect the authors to train a huge number of models on the dataset, but it would be appreciated if they provided more dataset-level statistics, especially with regard to the NSFW content.

For example, one piece of analysis that could improve the paper would be a human review of a small random subset of images (perhaps on the order of 10,000 images). This human review would help us assess the accuracy of the CLIP-based NSFW tagging (as well as the other models such as the watermark tagger). It may also identify some forms of harmful content which are not caught by the current filtering methodology. Relative to the high cost of data processing and model training, this would probably not be a large cost for the project overall.

A second example would be a more in-depth exploration of the different types of images contained in the dataset. For example, how many of these images are stock photos, how many are memes, how many are advertisements, etc. I would not expect exact numbers, but a broad estimate would still be valuable.

A third example would be a linguistic/NLP analysis of the captions. For example, how many of the captions are full sentences, how many contain proper nouns, how many contain numbers, how many contain locations, etc.

Some aspects of the data pipeline could also be enhanced to make the data more accessible and more widely-used. For example, the paper emphasizes in multiple places that it is designed for use primarily in an academic setting. However, as the paper also mentions, training models on data of this scale requires extremely large computational resources which are generally not available to academia (with the exception of a few supercomputers, such as JUWELS). Based on this, it seems likely that in practice, the dataset will be primarily used in industry and industrial research settings.

To facilitate use of the data in academic settings, it may be valuable to provide some tools for researchers with fewer computational/storage resources. For example, perhaps one could provide a utility for dataset streaming or a small but very high-quality subset of the data for academic use-cases.

**Additional Feedback:**

At least from this reviewer's perspective, the extent to which this extreme scale (i.e. 5B+ images) is necessary for training large-scale vision-language understanding is not yet clear. In other words, it is not clear from the paper that it is possible to achieve anything with 5B+ images that was impossible with 400M images. Additionally, if I read the paper correctly, I do not believe that the authors actually trained a model on the full LAION-5B dataset. Such a model would necessarily have to be multilingual (and may also be valuable to the multilingual NLP community), and there are no discussions of the performance of multilingual vision-language models.

For English-only tasks, Table 2 shows results for LAION-2B-en compared to LAION-400M and shows significant improvements across the board. However, it is not clear whether the improvements should be attributed to the dataset size or the additional training time/compute used to train the 5B-en model. Table 4 in the Appendix shows that the (ViT-B/32) 2B model was trained for 210 hours whereas the (ViT-B/32) 400M model was trained for 36 hours. If I am interpreting this correctly, the results are not comparable and they should not be presented in the same table as they are currently presented.

Indeed, Figure 4 shows that at the same level of compute, a (CLIP-ViT-B/16) model trained on 400M and (CLIP-ViT-B/32) model trained on LAION-2B-en deliver approximately the same downstream performance. It would be very helpful to have an apples-to-apples comparison of two models trained on the 2B and 400M subsets using the same set of hyperparameters and the same amount of total compute.

Additional small questions
 * The caption and description of Figure 4 mention a "log-log plot". Am I confused or is this figure a log-linear plot?

**Correctness:**

To the best of my knowledge, all claims made in the dataset are correct. The dataset pipeline is open-source and the quantitative results are reproducible using Open-CLIP.

**Documentation:**

The dataset and processing pipeline is open-source and well-documented.

**Ethics:**

Due to the web-scraped nature of this vast dataset, it will undoubtedly be replete with questionable content. I do not feel fully qualified to assess this work from an ethics and privacy perspective, and I recommend that there is a dedicated ethics/privacy review conducted by an expert. Nonetheless, I will try my best below.

1. General NSFW Content

The paper mentions very briefly that "after the release of LAION-400M, several groups investigated potential problems arising from an unfiltered dataset." These problems and criticisms merit more discussion than is given in the paper. It would be worth including a dedicated section in which you openly explain these points of contention and provide guidance about how to deal with issues of misogyny, pornography, and stereotypes in the data. It seems that this dataset paper has an obligation to discuss such issues openly, and this paper of the paper reads like something that is defensive rather than open to discussion.

Along these lines, it may be worth looking at multiple ways of tagging inappropriate content. Section 3.2 mentions that the two classifiers used for tagging are both CLIP-based; this may introduce some bias in the tagging/filtering that could be addressed with the use of other models or techniques. For example, CLIP may have difficulty detecting hateful memes or inappropriate text in images. Given the importance of this tagging to the behavior of downstream models, such issues merit thought and investigation.

In general, the text of the paper does its best to try to shift all responsibility for negative outcomes relating to the dataset from the dataset creators to the dataset users. The reviewer understands this behavior and is generally inclined to agree with this assignment of responsibility. However, the reviewer would understand if other reviewers or ethics experts requested more assurances about the potentially harmful parts of the dataset from the dataset creators.

Additionally, the authors should probably set the default filter settings (e.g. for LAION-2B-en) in a very conservative manner, such that it has as little undesirable content as possible.

2. Humans

There is only a small discussion in the paper about the presence of humans in the dataset. Regarding consent (Q32 in the dataset checklist), the paper states:
```
Users have full control over the presence of their data in our dataset. If users wish to revoke their consent, they can delete the underlying website – it will be automatically removed from LAION-5B since we distributed image-text pairs as URLs. Moreover, we provide a contact email to remove dataset samples: contact@laion.ai.
```
It strikes me as insincere to say that users have full control over their data because they can remove their images from the internet.

It would be good practice to describe how you intend to update the dataset when users revoke their consent to the use of their data.

3. Dataset Checklist

A few of the answers on the dataset checklist should be more forthcoming. For example, there is a question "Q20: Does the dataset contain data that might be considered sensitive in any way" and the answer is misleading. The answer should be a straight "yes" -- the dataset is large enough that it will absolutely contain some sensitive data.

**Relation To Prior Work:**

The dataset clearly discusses how it relates to prior work.

**Summary And Contributions:**

This paper introduces the LAION-5B dataset, an ultra-large-scale collection of image-text pairs designed for vision-language research. The dataset contains additional metadata about each image-text pair, including language information, resolution, and multiple content flags. The metadata enables researchers to filter the dataset into smaller sub-datasets, for example by language (LAION-2B-en) or resolution (LAION-High-Resolution).

The contribution of the paper is clear: the dataset is significantly larger than any previous image-text dataset openly available to the public. From the perspective of this reviewer, this is a significant and valuable contribution. There is little doubt that over the past five years, the increase in dataset scale has been an important factor in the large improvements of vision-language models. It seems likely that the trend of increasing model size, dataset size, and training time will continue, as it has in NLP.

Regarding explicit/questionable content, the authors leaned on the side of tagging and including data rather than removing it from the dataset. This decision enables researchers and practitioners to decide upon their own dataset filters, giving them the greatest amount of flexibility when constructing their data subset. On the other hand, the inclusion of a very large amount of explicit content could lead to some users training models on such content without intending to do so. Many organizations (especially those subject to European privacy laws) will inevitably have to re-filter that data in order to ensure that it complies with relevant policies and regulations. Further ethical/privacy issues are discussed in the ethics section below.

---

> ### Author Response · Authors · 2022-08-22
> **Author response, Part 1**
>
> We thank the reviewer for the positive rating and the many constructive points of feedback. We will now address the individual points:
>
>
> > The primary weakness of the paper is that there should be a more in-depth analysis of the dataset contents and filtering methodology.
>
> We agree that a more in-depth analysis of LAION-5B would be very interesting, and we hope to perform such analyses in future work. Our focus in this submission was to create the LAION-5B dataset, and to demonstrate that LAION-5B can be used for training large image-text models. Since this was already a substantial effort and there are numerous possible follow-up analyses, we decided that submitting our current set of results is hopefully already a useful starting point for the community to train models and conduct further analyses of the dataset.
>
> The reviewer may find our supplementary material interesting since it already contains some of the analyses the reviewer suggested. We provide specific pointers to the supplementary material in our responses below.
>
>
>
> > It is not reasonable to expect the authors to train a huge number of models on the dataset, but it would be appreciated if they provided more dataset-level statistics, especially with regard to the NSFW content.
> >
> > For example, one piece of analysis that could improve the paper would be a human review of a small random subset of images (perhaps on the order of 10,000 images). This human review would help us assess the accuracy of the CLIP-based NSFW tagging (as well as the other models such as the watermark tagger). It may also identify some forms of harmful content which are not caught by the current filtering methodology. Relative to the high cost of data processing and model training, this would probably not be a large cost for the project overall.
>
> About 3% of LAION was flagged as NSFW by our NSFW filter. We also agree that the analysis suggested by the reviewer is interesting and have in fact already conducted a similar experiment for our submission, as described in Appendix C.5 of the supplementary material. Specifically, we created a test set with 1,000 images from each NSFW category and manually inspected it to make sure that all test images were correctly annotated. Our EfficientNet-V2- B02 image classifier predicted 96.5% of the NSFW images correctly as NSFW and discarded 8.0% of the SFW images incorrectly as NSFW. We do not view our NSFW filter as final and plan to improve it in follow-up work.
>
>
>
> > A second example would be a more in-depth exploration of the different types of images contained in the dataset. For example, how many of these images are stock photos, how many are memes, how many are advertisements, etc. I would not expect exact numbers, but a broad estimate would still be valuable.
>
> We agree that this is an interesting question and plan to collect such statistics for the final version of the paper or future work. A related field that already exists in the LAION data is the “patermark” field from the watermark detector. See https://huggingface.co/datasets/laion/laion2B-en-joined and https://laion.ai/blog/laion-5b/ for details.
>
>
>
> > A third example would be a linguistic/NLP analysis of the captions. For example, how many of the captions are full sentences, how many contain proper nouns, how many contain numbers, how many contain locations, etc.
>
> Appendix D contains a first step into this direction. Specifically, Figure 9 shows the distribution of caption lengths, and Figure 10 shows the distribution over languages. We agree that more linguistic analyses would be interesting, but as mentioned above, we also consider a comprehensive linguistic analysis of LAION-5B to be beyond the scope of this paper. Our focus was on training large image-text models, whose performance we thoroughly validate.

---

> > ### Comment · Reviewer_XLgX · 2022-08-29
> > **Thank you for your comprehensive response**
> >
> > I would like to thank the authors for their very thorough response to my questions and comments.
> >
> > The authors have done a very good job of addressing my concerns. The authors acknowledge that much of this work is a first step toward large open image-text datasets and all points they make are reasonable. I appreciate the figure with experiments conducted with various fractions of LAION-400m data and the author's reference to the findings of the BASIC paper. I am looking forward to future analysis of this data.
> >
> > My primary concerns were related to ethics and I was glad to see that this paper received an ethics review. I thought the author's response to the ethics review and my ethics-related concerns was quite comprehensive. Given that ethics reviewer was satisfied, I am also satisfied on this front.
> >
> > Finally, it is important to mention the recent release and adoption of Stable Diffusion, which was of course trained on this dataset. This model demonstrates the impact that this dataset can make in an open/academia-like setting; the dataset is clearly a valuable resource for the community and will see widespread adoption.
> >
> > On the basis of these points, I will be raising my score to 8. I highly recommend acceptance.

---

> ### Author Response · Authors · 2022-08-22
> **Author response, Part 2**
>
> > Some aspects of the data pipeline could also be enhanced to make the data more accessible and more widely-used. For example, the paper emphasizes in multiple places that it is designed for use primarily in an academic setting. However, as the paper also mentions, training models on data of this scale requires extremely large computational resources which are generally not available to academia (with the exception of a few supercomputers, such as JUWELS). Based on this, it seems likely that in practice, the dataset will be primarily used in industry and industrial research settings.
>
> Regarding accessibility of our pipeline, it is worth mentioning that we have already released the individual building blocks as open source software tools. For instance, the img2dataset repository has already received more than 800 stars on GitHub: https://github.com/rom1504/img2dataset. In addition, we plan to develop a streamable version of img2dataset to be used as a custom dataloader, e.g., in PyTorch, that will allow users with limited disk space to use our dataset.
>
> We would also like to point out that some academic groups have already built on the LAION dataset despite its size. For instance, the Machine Vision and Learning research group at the Ludwig Maximilian University of Munich has utilized LAION to build large-scale diffusion models: https://github.com/CompVis/stable-diffusion. As another example, the following paper from the University of Washington analyzes the robustness properties of LAION compared to other data sources: https://arxiv.org/abs/2208.05516. Hence we are optimistic that academic researchers, sometimes in collaboration with companies, will be able to utilize LAION productively in their research.
>
>
>
> > To facilitate use of the data in academic settings, it may be valuable to provide some tools for researchers with fewer computational/storage resources. For example, perhaps one could provide a utility for dataset streaming or a small but very high-quality subset of the data for academic use-cases.
>
> There are already small, highly curated image-text datasets such as CC3M and CC12M, see https://github.com/google-research-datasets/conceptual-captions and https://github.com/google-research-datasets/conceptual-12m . Hence our focus was primarily on large-scale datasets. Nevertheless, we have already released small, specialized subsets of LAION such as LAION-art and LAION-aesthetic, see: https://github.com/LAION-AI/laion-datasets/blob/main/laion-aesthetic.md We plan to build more specialized high-quality subsets in the future.
>
>
>
> > At least from this reviewer's perspective, the extent to which this extreme scale (i.e. 5B+ images) is necessary for training large-scale vision-language understanding is not yet clear. In other words, it is not clear from the paper that it is possible to achieve anything with 5B+ images that was impossible with 400M images. Additionally, if I read the paper correctly, I do not believe that the authors actually trained a model on the full LAION-5B dataset. Such a model would necessarily have to be multilingual (and may also be valuable to the multilingual NLP community), and there are no discussions of the performance of multilingual vision-language models.
>
> The BASIC paper (https://arxiv.org/abs/2111.10050) already demonstrated convincingly that the extreme scale of 5B+ images can substantially increase the performance of vision-language models. Specifically, BASIC was trained on 6.6 billion image-text pairs and achieved about ten percentage point higher ImageNet zero-shot performance than OpenAI’s largest CLIP model trained on 400 million images (see Table 1 in the BASIC paper).
>
> Regarding multi-lingual CLIP models, we agree that this research direction is still in its infancy. We are planning to explore this direction more in the future, building on LAION-5B and our collaboration with https://github.com/FreddeFrallan/Multilingual-CLIP .

---

> ### Author Response · Authors · 2022-08-22
> **Author response, Part 3**
>
> > For English-only tasks, Table 2 shows results for LAION-2B-en compared to LAION-400M and shows significant improvements across the board. However, it is not clear whether the improvements should be attributed to the dataset size or the additional training time/compute used to train the 5B-en model. Table 4 in the Appendix shows that the (ViT-B/32) 2B model was trained for 210 hours whereas the (ViT-B/32) 400M model was trained for 36 hours. If I am interpreting this correctly, the results are not comparable and they should not be presented in the same table as they are currently presented.
> >
> > Indeed, Figure 4 shows that at the same level of compute, a (CLIP-ViT-B/16) model trained on 400M and (CLIP-ViT-B/32) model trained on LAION-2B-en deliver approximately the same downstream performance. It would be very helpful to have an apples-to-apples comparison of two models trained on the 2B and 400M subsets using the same set of hyperparameters and the same amount of total compute.
>
> We thank the reviewer for this important question, which we also discuss in our response to Reviewer hzjq. Due to compute limitations, we could not yet run all experiments to investigate what happens if LAION-400m training is performed with the same compute budget as LAION-2B training.
>
> As a first step in this direction, we have carried out an additional experiment where we train CLIP models on different subsets of LAION-400m. In particular, we trained an openCLIP ViT B/32 model on a subset of size 80 million images drawn from LAION-400m, where we used the same total compute budget as for training on 400 million images (160 epochs for 80m to match 32 epochs used for LAION-400m). The downstream zero-shot accuracy of the resulting model is still substantially below training with 400 million images in this case (54.89% for B/32 on 80m (160 epochs) vs 62.9% on LAION-400m (32 epochs). Here we provide the figure showing experiments conducted with various fractions of LAION-400m data, including the 80m experiment with extended 160 epochs budget:
>
> https://media.discordapp.net/attachments/947179319267065977/986213724660592680/data_vs_accuracy.png
>
> This provides evidence that data scale matters. Even with increased compute, the model trained with the smaller 80m training set appears bottlenecked by the data scale and stays below the model trained on the 5x larger 400m training set (with an equal amount of compute).
>
> Currently we are also in the process of running the experiment the reviewer suggested with a B/32 model, training it on LAION-400m for 80 epochs (which corresponds to 16 epochs used for B/32 training on LAION-2B). We hope to be able to report the results in the final manuscript, but due to the scale of the experiment we unfortunately cannot promise this.
>
>
>
> > The caption and description of Figure 4 mention a "log-log plot". Am I confused or is this figure a log-linear plot?
>
> This is indeed a log-linear plot - thank you for pointing out this typo!
>
>
>
> Finally, we would like to thank the reviewer for their in-depth comments regarding the ethics of our submission. We will improve our manuscript as suggested by the reviewer and provide a more detailed response to the ethics points together with our response to the overall ethics review.

---

### Official Review · Reviewer_DgCg · 2022-07-27
**An interesting large-scale dataset for multi-modal research**

**Rating:** 8
**Confidence:** 4
**Clarity:** Yes.

**Strengths:**

(1) The proposed LAION-5B is the largest openly available dataset for training vision-and-language models.

(2) The data collection and curation process has been comprehensively discussed to support community investigation and further improvement.

(3) Experiments support that CLIP models trained on LAION-400M match the performance of original CLIP models trained on the private dataset.

(4) The dataset will add much value to the multi-modal learning community, regarding 1) being able to train large-scale models, 2) publishing pre-trained models for many downstream tasks, and 3) being able to audit and refine a dataset of this magnitude.


**Weaknesses:**

(1) Why does the CLIP model with ViT-L/14 trained on LAION-400M perform largely worse than the original CLIP model with ViT-L/14, compared to the cases with ViT-B/32 and ViT-B16?

(2) In the abstract, the authors claimed that “we show successful replication and fine-tuning of foundational models like CLIP and GLIDE using the dataset”. However, I didn’t see the experiments of replication or fine-tuning on GLIDE models.

(3) Figure 2 is difficult to interpret. Maybe add more detailed caption descriptions and also refine the whole diagram to make it more accessible.

(4) What are the WAT files in Common Crawl?

(5) In the Related Work section, I suggest putting the reference numbers before the full stop at the end of a sentence.


**Additional Feedback:**

N/A

**Correctness:**

The benchmark is constructed in a sound way and the evaluation methods and experiment design well support the claims about replicating CLIP. However, I didn’t see experiments to support the claims about replicating GLIDE.


**Documentation:**

Yes.

**Ethics:**

The authors have well discussed the ethical issues the dataset might have.

**Relation To Prior Work:**

Yes.

**Summary And Contributions:**

This paper created a new open dataset LAION-5B that contains over 5.85 billion image-text pairs, of which 2.32 billion contain English language. Besides, this work provided a replicable pipeline for the data curation process, based on Common Crawl, including data acquisition, distributed processing of Common Crawl, distributed downloading of images, and post-processing.  To validate the usefulness of LAION-5B in training and analyzing the large-scale vision-and-language models, this work trained the CLIP models of various scales on a subset of the dataset and showed that they match the strong zero-shot and robustness performance of the original CLIP counterparts trained on closed curated data. Finally, the safety, ethical concerns, and data biases were thoroughly discussed.

---

> ### Author Response · Authors · 2022-08-22
> **Review response**
>
> We thank the reviewer for the positive feedback and recognizing the value of our dataset. We now address the weaknesses pointed out by the reviewer:
>
> > (1) Why does the CLIP model with ViT-L/14 trained on LAION-400M perform largely worse than the original CLIP model with ViT-L/14, compared to the cases with ViT-B/32 and ViT-B16?
>
> The 3 percentage point (pp) accuracy drop of a LAION-trained ViT-L compared to an OpenAI-trained ViT-L is indeed an interesting phenomenon. We conjecture that the accuracy drop stems from one of the following three causes:
>
> A) The CLIP-based filtering in the LAION dataset curation process, where we employed only a small scale ViT-B/32 model, which may have resulted in a substantial amount of poorly matching image-text pairs (please also refer to our response to reviewer CS2n).
>
> B) The different source set: Common Crawl for LAION and unknown data sources for OpenAI’s training set. Common Crawl may be a more noisy data source (weaker connection between images and associated text) or contain less diverse images.
>
> C) Different hyperparameter choices when training our CLIP models (e.g., the batch size and learning rate schedule).
>
> In this context, it is worth noting that our LAION-trained ViT-L model is still 4.5 pp better than OpenAI’s ViT-B/32 model we used for filtering LAION. So despite filtering with OpenAI’s ViT-B/32 model, the LAION dataset still enables accuracy increases beyond the accuracy of OpenAI’s ViT-B/32 model by increasing the model size of the model we train.
>
> For completeness, we would also like to point out that not only the ViT-L model sees an accuracy drop when trained on LAION (compared to OpenAI’s model). As we show in Table 2, the zero-shot ImageNet accuracy of a ViT-B/16 model trained on LAION is also 1.3 pp lower than a ViT-B/16 model trained by OpenAI. So overall LAION seems to match the accuracy at the model scale that was used for filtering (a ViT-B/32 model), and then sees increasing accuracy gaps to the OpenAI models as we increase model scale. This trend indicates that using a higher accuracy model for filtering LAION is a promising direction for future work to improve our dataset.
>
> In follow-up work, we intend to create a CLIP ViT L/14 filtered version of LAION-400m and test if this will further improve results obtained when training on a larger scale, specifically B/16 and L/14 (please also refer to our response to reviewer CS2n). We hope that this may close the gap observed in the current work. Furthermore, we have ongoing experiments that suggest that we may be able to close the accuracy gap already for LAION-400m by changing the training hyperparameters. Due to the computational scale of these experiments, they will unfortunately not finish before the discussion period ends.
>
>
>
> > (2) In the abstract, the authors claimed that “we show successful replication and fine-tuning of foundational models like CLIP and GLIDE using the dataset”. However, I didn’t see the experiments of replication or fine-tuning on GLIDE models.
>
>
> We agree that work on GLIDE could have been covered better and will include a subsection on our GLIDE reproduction - LAIONIDE - in the supplementary material (see https://replicate.com/laion-ai/laionide-v2). LAIONIDE is a GLIDE based model trained on subsets of LAION-5B and other datasets. For a comparison to OpenAI GLIDE, please see the example images on the following page: https://wandb.ai/afiaka87/laionide-v2-coco-eval/reports/Qualitative-evaluation-between-OpenAI-GLIDE-and-Laionide-V2---VmlldzoyNDkwNDUz.
>
>
>
> > (3) Figure 2 is difficult to interpret. Maybe add more detailed caption descriptions and also refine the whole diagram to make it more accessible.
>
> We have already improved Figure 2 after submitting our paper and will include the following updated version in the paper: https://drive.google.com/file/d/1QPWVQv0xKXb18A_0_HH-Cn_ALy5-SGz-/view?usp=sharing
> Any additional comments on this updated version of Figure 2 are welcome.
>
>
> > (4) What are the WAT files in Common Crawl?
>
> In the context of Common Crawl, WAT files are metadata files. Please see https://commoncrawl.org/the-data/get-started/ for further documentation. We will clarify this in the updated version of our paper and include this link to the Common Crawl documentation.
>
>
> > (5) In the Related Work section, I suggest putting the reference numbers before the full stop at the end of a sentence.
>
> We agree with this formatting suggestion and will update our paper accordingly.

---

> > ### Comment · Reviewer_DgCg · 2022-08-29
> > **Thanks for your response**
> >
> > Thank the authors for providing a detailed response. My major concerns have been well addressed. I hope to see the updated results in the revised paper, particularly including the CLIP ViT L/14 filtered data and results, and LAIONIDE. Thus, I increase my score to 8 and recommend a clear acceptance.

---

### Official Review · Reviewer_pcKV · 2022-07-28
**Exceptional large scale dataset with high quality analysis, performance, and review.**

**Rating:** 8
**Confidence:** 4

**Strengths:**

1. The dataset describes and utilizes a very well defined aggregation methodology.
2. The paper and dataset employ a useful approach of cosine similarity filtering, employing CLIP to compute these scores.
3. The authors apply several safeguards to prevent unsavory and biased information from affecting the dataset. The application of these safeguards does not, in itself, resolve all issues, but this is a step in the correct direction.
4. The authors evaluate the models trained on their dataset using a variety of tasks and provide detailed information in the supplementary material regarding these experiments.
5. Ethical concerns have discussion that is equal to that of its predecessor papers.
6. Allowing access to all data enables independent probing and review of the dataset itself.
7. Technical limitations are well outlined and clearly explained.
8. The supplementary materials (webpage, blog, services, and figures) are a true highlight. This meets the high-water-mark for excellent documentation and supplemental information.

**Weaknesses:**

This paper is certainly significant within the field, however there are some important weaknesses to mention.
1. Investigation of other model architectures and their scaling to a dataset of this size would be useful to include.
2. The inclusion of few-shot learning or a linear probe for downstream evaluation tasks would allow for more comparisons to other datasets.
3. GLIDE and generative works could have been attached in the supplementary or explained in greater detail.
4. While there is a reasonable amount of ethical review, it would be nice to see a greater probe into the nature of the content throughout LAION 5B. There could have been more manual analyses performed. Furthermore, while the authors attempt to mitigate the risk of potential harms in their dataset, the certainty with which they assert their mitigations could lead to confusion. This point is continued in the ethics section.


**Additional Feedback:**

Great paper, I think supplementary investigations into the dataset as a whole as well as a deeper look into potential ethical hazards would make a great follow up.

**Clarity:**

The paper is very well written and clearly explained. Particular kudos to the authors as the paper seems easily understandable to even a non-technical audience. This is increasingly important as this paper will likely be cited, paraphrased, and consumed by a different audience than this conference. There are minor typos in the paper (eg. the trailing hyphen in this line"one prominent example is data efficient zero- and few-shot c"), but overall the paper is extremely well written. The supplementary sections as well are very clear.

**Correctness:**

The dataset creation, testing, and evaluation methodology are all sound. This paper appears correct about its claims. The supplementary section also answers questions correctly.

**Documentation:**

The webpage, blog, viewer, and supplementary materials are all exceptional. It would be nice to see more manual reviews or even explorations into the dataset presented, but the time complexity of engaging in that workload is understandable. Either way, as mentioned in the strenghts, the documentation is incredibly well done.

**Ethics:**

It is first important to state that this review functions as a pragmatic look into the ethics of this paper. This field has growing issues with ethical hazards that become increasingly difficult to both understand and mitigate, particularly at the scale of this dataset. Even a manual review of images and text can still result in the leakage of potentially hazardous content (as seen for example in this paper https://openreview.net/pdf?id=HJlrwcP9DB). However, it is also important to understand the context surrounding a dataset's creation. If the authors demonstrate an intention to mitigate potential harms, as well as an understanding of how ethical hazards could propagate into models trained on this dataset, then this paper clears current ethical barriers.

As a personal note, I strongly believe that there should be more oversight into the data that is used to train large scale machine learning models, particularly as our manual capability to evaluate and probe both models and datasets diminishes relative to the scale of the aforementioned datasets and models. To mitigate the bias this belief contributes to my review, I solely compared this paper to the Neurips Ethical Guidelines, and the relevant ethical literature in the field.

Overall, I find the level of ethical consideration in this paper sufficient but unsatisfactory. By clearly outlining the terms of use for the dataset, as well as employing state-of-the-art methods to filter and remove unsavory content from both the image and text portions of the dataset, this dataset maintains the current standard for ethical filtering. The method of using the CLIP embedding space to then train a model for NSFW detection, evaluated on a supervised dataset of only 1000 examples could be bolstered by comparisons with other NSFW detection techniques. Subsequently, a deeper analysis into the usage of certain vernacular that is more popular across different demographic groups, and how the filtering steps affect the prevalence of this content would prove extremely valuable.

The datasheet attached to this paper is answered sufficiently and correctly. This allows for easy comparison with other datasets that provide datasheets and is extremely helpful.

The authors clearly understand the ethical issues that affect this dataset, one of which is the bias introduced by using CLIP as a filtering source. Further investigation of how this bias affects the data within LAION 5B would prove extremely insightful, but is not provided. Again, this is not a failure of the paper, but rather an important ethical issue that warrants both discussion and investigation.

Ultimately, this paper continues the current standard of ethical literature surrounding datasets. This field has progressed significantly in the last few years in regards to ethical safeguards, but the lack of detailed analysis and investigation in this paper prove that there is still a long way to go. However, the sheer exposure of a dataset in this scale to the research community is highly valuable (as datasets of this scale or larger already exist to train private models for corporations) and will certainly bolster future ethical research. The answers and investigations of the authors of this dataset prove sufficient to meet the current expectations for published datasets, but they are ultimately unsatisfactory as the authors rarely contended with the unique scale of their data, nor did they extend the field as a result of their unique dataset. Unsatisfactory in this context only implies that the paper could have done more for the field, but does not affect the validity of the ethical discussion in this paper.

TL;DR
The paper keeps pace with what is expected of published datasets, but it should grapple with the uniquely large scale and also attempt to extend the discussion of dataset ethics.

**Relation To Prior Work:**

This paper does discuss its relation to prior work quite extensively. Both throughout the paper and in the related work section, the paper continually references related work.

**Summary And Contributions:**

The authors propose a massive 5+ billion instance dataset comprised of image-text pairs. This functions as a novel exceptionally large dataset for the research community, and is the only open-sourced one of its size. On top of its large size, LAION 5B demonstrates novel data aggregation techniques, sophisticated curation pipelines, and extends several key analyses. By providing both a reproducible process and pipeline, as well as open sourcing all their data and code, LAION 5B exhibits a high quality standard of reproducibility and visibility. Furthermore, LAION 5B provides access to a large open source implementation of CLIP. By collecting images using CLIP as a filter, and adequately screening for ethical and moral hazard (while providing important disclosures about this dataset's usage), LAION continues the work of other papers demonstrating that important data quality analysis can be performed at large scales.

---

> ### Author Response · Authors · 2022-08-22
> **Review response, Part 1**
>
> We thank the reviewer for the positive review. We would like to address main points raised by the reviewer as follows:
>
> > Investigation of other model architectures and their scaling to a dataset of this size would be useful to include.
>
> We agree that this is a very interesting direction for future work. However, such an investigation exploring more architectures would have exceeded our computational budget, which was already substantial for a community-driven effort (340,000 GPU hours on the JUWELS supercomputer). Hence we decided to defer such an investigation to future research and focus on OpenAI’s original CLIP architectures as these are the best to validate LAION-5B.
>
> More specifically: OpenAI’s CLIP models are currently the most widely used image-text models and their performance is comparatively well understood. Follow-up work introducing new architectures building on the original CLIP models often have one of the following limitations: (i) smaller models, (ii) trained on smaller datasets, and / or (iii) less comprehensive evaluation than OpenAI’s CLIP paper. In our experiments, reported performance improvements at small scale often did not transfer to performance improvements at the larger scale (400M+ training images) we are interested in. Furthermore, those architectures were evaluation was perfomed at larger model and /or data scales (eg., ALIGN using ca. 1.8B image-text pairs or BASIC using ca. 6B non-public datasets for training) do not report results from intermediate scales and do not  provide corresponding pre-trained models that would make systematic comparison in our frame possible. Overall, this made the original CLIP models, which are thoroughly validated at large scale already, providing pre-trained models for a range of scales with data scale matching to ours, a natural first step for our reproduction effort.
>
>
> > The inclusion of few-shot learning or a linear probe for downstream evaluation tasks would allow for more comparisons to other datasets.
>
> We thank the reviewer for the good suggestion. Reporting zero-shot results on ImageNet, VTAB, and robustness datasets was our first step towards validating LAION-5B for large-scale multi-modal training. Linear probing and few-shot fine-tuning results would further substantiate the comparison to OpenAI’s original CLIP models. To extend our comparison, we performed further few-shot fine-tuning experiments on ImageNet, also evaluating robustness of resulting few-shot fine-tuned models. The results are as following:
>
> Few-shot transfer performance, comparison openCLIP and original openAI CLIP:
> [PDF Figure Few Shot Transfer](https://drive.google.com/file/d/1FdeyNfutK5ztRAN8_2E3AOuhT6218Utj/view?usp=sharing)
>
> Average accuracy of few-shot transfer models on 5 different robustness datasets:
> [PDF Figure Few Shot Robustness](https://drive.google.com/file/d/1tOpCkWsvXY7sBNQGdr7PPzULhPfD6LrW/view?usp=sharing)
>
> In line with our findings on zero-shot transfer performance, we observed that while our openCLIP ViT B/32 and B/16 model are matching very closely the results reported from the original CLIP paper, openCLIP ViT L/14 shows again a systematic gap across few shot conditions. As discussed before, we assume that the gap on the larger scale may be due to the filtering of LAION performed by smaller scale pre-trained original CLIP B/32 model.
>
> Again we see the positive effect of model and data scale on measured transfer performance in few-shot scenario - both increasing network size (B/32, B/16, L/14) and data scale (B/32 on LAION-400m vs B32 on LAION-2B) leads to significantly improved transfer performance across all few-shot conditions. We conclude that evaluating few-shot transfer performance further validates LAION as a dataset for pre-training strong transferable and robust vision-language models at scale.  We will include the figures of the performed few-shot transfer experiments in the Appendix of the final manuscript version.
>
> > GLIDE and generative works could have been attached in the supplementary or explained in greater detail.
>
> We agree that work on GLIDE could have been covered better and will include a subsection on our GLIDE reproduction - LAIONIDE - in the supplementary material (see https://replicate.com/laion-ai/laionide-v2). LAIONIDE is a GLIDE based model trained on subsets of LAION-5B alone (v2) and additional other datasets (v3). For a comparison to OpenAI GLIDE trained solely on LAION-5B subsets, please see the example images on the following page: https://wandb.ai/afiaka87/laionide-v2-coco-eval/reports/Qualitative-evaluation-between-OpenAI-GLIDE-and-Laionide-V2---VmlldzoyNDkwNDUz.
>
> In addition, we will update our related work section to include new generative models that have appeared in the meantime and build on LAION-5B, for instance a new diffusion model hosted by stability AI: https://stability.ai/blog/stable-diffusion-announcement  (based on https://github.com/CompVis/stable-diffusion)

---

> ### Author Response · Authors · 2022-08-22
> **Review response, Part 2**
>
>
> > While there is a reasonable amount of ethical review, it would be nice to see a greater probe into the nature of the content throughout LAION 5B. There could have been more manual analyses performed. Furthermore, while the authors attempt to mitigate the risk of potential harms in their dataset, the certainty with which they assert their mitigations could lead to confusion. This point is continued in the ethics section.
>
> We appreciate the reviewer’s thoughtful comments regarding the ethical aspect of our work. We also grappled with these questions when preparing our submission and are looking forward to additional feedback from the ethics review. We certainly agree that more can be done to assess the ethical implications of our dataset and mitigate potential problems. As we describe in our submission, we believe that we have taken effective first steps and hope that the community will join us in future work to improve LAION from both an ethical and technical perspective. We will respond to ethical concerns in more detail in the available ethics review (which also raised some similar points, with general overall conclusion that current work has no serious ethical issues). We will also revise the discussion sections in our paper to clarify that our mitigations cannot address all potential ethical concerns.
>
> Finally, we thank the reviewer for their attention to detail and pointing out the typo in the line "one prominent example is data efficient zero- and few-shot c". We would appreciate any additional feedback regarding our writing and will also proofread our paper again ourselves to improve the writing.

---

### Official Review · Reviewer_CS2n · 2022-07-30
**The review**

**Rating:** 8
**Confidence:** 4
**Correctness:** Yes. It is constructed in a sound way.
**Clarity:** Yes. It is easy to follow.

**Strengths:**

A good paper.

It is very happy to see such a big dataset (5 billion image-text pairs) open-sourced since this can significantly boost the development of language-vision pretraining techniques.

The image-text pairs in the dataset are divided according to the languages, and each part is on a large scale. This provides new opportunities for multilingual and potentially low-resource language researchers.

The pairs are well-tagged with toxic, offensive, and pornographic. This encourages research in ﬁelds such as dataset curation.

Several research works have been conducted based on a pre-released subset LAION-400M, which proves the value of the proposed dataset.

The potential bias caused by such a large dataset is well discussed in the paper.

**Weaknesses:**

1) Figure 3 is missing.

2) The data is filtered by ViT-B/32 in the post-processing part. Is there any chance that the data would be filtered by a much larger model like ViT-L? Besides, as the author claimed that the CLIP model in itself introduces a bias towards LAION, are there any other techniques in the future to avoid this bias and even measure the effect brought by the bias? It would be appreciated if the authors can discuss it.

**Additional Feedback:**

A good paper. I think it is far above the acceptance bar thus I would give a rating of 8 for this paper.

**Documentation:**

Yes.

**Ethics:**

Yes.

**Relation To Prior Work:**

Yes

**Summary And Contributions:**

This paper proposed the first publically available large-scale dataset of image-text pairs. It consists of 5.8 billion pairs in total and the authors successfully reproduced the CLIP models of different sizes using the proposed dataset. The toxic, offensive and pornographic content are tagged rather than removed to encourage research in ﬁelds such as dataset curation. The data bias and limitation is well discussed and notified in the paper. Extensive experiments validate the effectiveness of the proposed paper.

---

> ### Author Response · Authors · 2022-08-22
> **Review response**
>
> We thank the reviewer for the positive feedback. We respond to the two main points raised by the reviewer below:
>
> > Figure 3 is missing.
>
> We apologize for this oversight when preparing our PDF for main submission. There was a formatting error while compiling the main document that escaped our attention. We already attached Figure 3 to the supplementary material, both as a separate image file in the zip archive and as Figure 7 in the supplement PDF. We will also update our paper to include the Figure 3 in its right place in the final version.
>
>
> > The data is filtered by ViT-B/32 in the post-processing part. Is there any chance that the data would be filtered by a much larger model like ViT-L? Besides, as the author claimed that the CLIP model in itself introduces a bias towards LAION, are there any other techniques in the future to avoid this bias and even measure the effect brought by the bias? It would be appreciated if the authors can discuss it.
>
> We agree that filtering the Common Crawl candidate pool with other models is a very interesting direction for future research. Due to the cost of training models on LAION, we did not pursue this direction in our paper since we would not have been able to train models on different versions of LAION.
>
> We conjecture that filtering Common Crawl with a CLIP ViT-L model will further increase the quality of our dataset. We decided to use a ViT-B/32 model for all filtering because OpenAI did not initially release larger CLIP models. Since we began the LAION data collection with a ViT-B/32 model, we decided to stick to this model for the entire dataset to maintain a consistent filtering pipeline even after OpenAI released larger CLIP models.
>
> For the follow-up work, we intend to create a CLIP ViT L/14 filtered version of LAION-400m to test how this affects model training and downstream transfer performance. We think that filtering by a small scale CLIP ViT-B/32 may leave more image-text pairs with weak or no semantic connection in the dataset while also accidentally removing some high quality image-text pairs. The larger CLIP ViT-L/14 model may thus create a less noisy version of LAION than what was possible with smaller scale CLIP ViT-B/32. This in turn will then hopefully improve the downstream transfer performance of models pre-trained on the ViT-L filtered version of LAION.
>
> A first step for measuring the biases induced by the CLIP filtering in LAION is to evaluate a broad range of downstream tasks, as we do in our submission. The next step would be to evaluate on more downstream tasks and assemble test sets specifically created to highlight biases induced by CLIP filtering (e.g., testing knowledge of events that happened after the OpenAI CLIP models were released). Once we have measured the biases induced by CLIP filtering more precisely, we hope we can mitigate biases by appropriately modifying the training data in LAION-5B.

---

> > ### Comment · Reviewer_CS2n · 2022-08-29
> > **The response**
> >
> > Thank you for taking my comments into consideration for the follow-up work. I believe that CLIP ViT L/14 filtered version of LAION-400m can have higher quality, which could benefit the performance on a lot of downstream tasks. My concerns have been well addressed.

---

### Review · Ethics_Reviewer_zKCR · 2022-08-20

**Recommendation:** 1

**Ethics Review:**

All technical reviewers give this paper very strong scores, and clearly view it as a substantial research contribution. The authors have also clearly put some effort into adhering to ethical best practices (e.g., adding annotations for offensive content, adding a datasheet). I agree with the reviewers that this paper should be accepted; I would suggest some changes in the final version though.

1. Language around IRB seems non-standard. "Since the data is found on the internet it is reasonably public, we did not need to consult an institutional review board." Generally, researchers aren't supposed to make the determination themselves about whether they should consult IRB. More importantly, "the data is on the internet so no IRB is needed" is not, in general, true.
2.   "Users have full control over the presence of their data in our database" ~ I agree with the reviewer who says this is a bit misleading. To remove their data from the database, users would have to know the database existed, have the technical literacy to download the dataset + search for their data, etc. In practice, lots of folks who did not consent to inclusion in any meaningful way are going to be scooped up by this dataset.
3. Beyond these specific comments, I encourage the authors to address the technical reviewers' thoughtful and detailed ethical comments, which I broadly agree with.

I am conflicted about this paper. It is clearly an impressive technical achievement. It is equally clearly dual-use (i.e., able to be used for good and bad uses). As an AI ethics researcher, I agree that it will be extremely useful for beneficial purposes - studying bias, inappropriate content, etc. At the same time, it can easily be put to bad uses (one could train a model just on the inappropriate content) or, more plausibly, negligent uses (I doubt very much that the researchers' justified warning to "not use the data in production" will actually keep it from being used that way by careless teams). I hope our community will be wise enough to do more good than bad with resources like this, and that we continue discussing best practices for releasing them. (Among other things, I don't think ethics reviews are really sufficient to keep harmful datasets from getting out into the world - they'll still be posted, cited, and used.)

---

> ### Author Response · Authors · 2022-08-28
> **Ethics response - part 1 (response to the ethics review, first part)**
>
> We have divided our response to ethics-related points into two parts. First, we will address the ethics review. Then we will respond to the ethics concerns raised by the other reviewers.
>
> # Ethics review
>
> We thank the reviewer for their constructive feedback on the ethical issues around our work. We are glad that the reviewer finds our work free of serious ethical issues, while pointing out caveats we have not treated to full extent. In following, we would like to respond to the main points raised by the reviewer:
>
> > Language around IRB seems non-standard. "Since the data is found on the internet it is reasonably public, we did not need to consult an institutional review board." Generally, researchers aren't supposed to make the determination themselves about whether they should consult IRB. More importantly, "the data is on the internet so no IRB is needed" is not, in general, true.
>
> We thank the reviewer for pointing this out and will consult the IRB from one of our institutions to get official guidance.
>
> > "Users have full control over the presence of their data in our database" ~ I agree with the reviewer who says this is a bit misleading. To remove their data from the database, users would have to know the database existed, have the technical literacy to download the dataset + search for their data, etc. In practice, lots of folks who did not consent to inclusion in any meaningful way are going to be scooped up by this dataset.
>
> We agree that removing oneself from LAION is above the technical level of most Internet users, as Reviewer XLgX also pointed out. We will update our datasheet accordingly. In addition, we plan to take the following steps to reduce how many image-text models are trained on identifiable humans in LAION:
>
> First, we plan to release human and face detections with LAION. Concretely, we have already run a pre-trained YOLO-v4 model and detected about 99 million humans with bounding boxes in LAION-2B-en.
>
> Building on the YOLO-v4 detections, we plan to release a version of LAION with no humans in the training set. We hope that this version of LAION will be useful for scenarios where the resulting model will not be applied on images containing humans.
>
> Following the example of face obfuscation in ImageNet (https://arxiv.org/abs/2103.06191), we also plan to adapt their code to work on LAION. This will enable users of LAION to train on images where humans are still present, but obfuscated. In addition, face inpainting methods may be able to replace the faces in LAION with generated ones if we can find inpainting methods with sufficient privacy guarantees (https://arxiv.org/abs/2005.09544 may be a candidate).
>
> None of these approaches will make it impossible to train on identifiable humans because a model developer utilizing LAION can still access the original images without any modifications. However, it is also worth noting that we ultimately cannot prevent model developers from utilizing this data: since all of the data in LAION is publicly available as part of Common Crawl, a sufficiently motivated adversarial model developer could always collect data directly from Common Crawl. Hence we can only make it as easy as possible to avoid training on identifiably humans for a model developer who is cooperative and wishes to do so.
>
> > Beyond these specific comments, I encourage the authors to address the technical reviewers' thoughtful and detailed ethical comments, which I broadly agree with.
>
> We respond to the ethics-related comments of the other reviewers in the second half of our ethics response below.

---

> ### Author Response · Authors · 2022-08-28
> **Ethics response - part 2 (response to the ethics review, second part)**
>
> > I am conflicted about this paper. It is clearly an impressive technical achievement. It is equally clearly dual-use (i.e., able to be used for good and bad uses). As an AI ethics researcher, I agree that it will be extremely useful for beneficial purposes - studying bias, inappropriate content, etc. At the same time, it can easily be put to bad uses (one could train a model just on the inappropriate content) or, more plausibly, negligent uses (I doubt very much that the researchers' justified warning to "not use the data in production" will actually keep it from being used that way by careless teams). I hope our community will be wise enough to do more good than bad with resources like this, and that we continue discussing best practices for releasing them. (Among other things, I don't think ethics reviews are really sufficient to keep harmful datasets from getting out into the world - they'll still be posted, cited, and used.)
>
> We thank the reviewer for appreciating our technical contribution. We also agree that LAION is an example of the dual-use dilemma. When deciding to release our dataset, we considered two kinds of potential negative consequences (apart from the beneficial purposes LAION can hopefully be used for):
>
> A) Enabling actively harmful actors.
>
> B) Enabling negligent behavior by careless teams.
>
> Regarding A), we think it is unlikely that LAION-5B changes the landscape substantially. Training an image-text model on our dataset requires large computational resources. Any organization with such resources is likely able to extract many images from Common Crawl (LAION’s source dataset) themselves, because Common Crawl is an existing public dataset.
>
> Moreover, the existence of public CLIP-style image-text models trained on LAION also does not enable actively harmful actors in a new way. Even the largest variants of OpenAI’s CLIP models have been publicly available as model checkpoints for several months now. OpenAI initially released only small CLIP models publicly, and then released larger CLIP models over time because there were no noteworthy negative applications of CLIP.
>
> Regarding B), we hope that a public, shared dataset scrutinized by the research community may actually help mitigate this kind of harm. As the reviewer noted, we also conjecture that product teams in industry will not actively want to cause harm, but will instead focus on company objectives and not devote enough resources to auditing their datasets for biases. If such teams build their own in-house dataset, the resulting datasets will likely not be audited by anyone. If such teams instead use LAION, there is at least a chance that they will benefit from community efforts to mitigate biases in this public dataset.
>
> Ultimately we share the same hope as the reviewer that our community will do more harm than good with the LAION dataset. And while we agree that ethics reviews cannot prevent harmful datasets from appearing, we still found the feedback we received as part of this process very helpful and believe that it will improve both our paper and the dataset.

---

> ### Author Response · Authors · 2022-08-28
> **Ethics response - part 3 (responses to the ethics points raised by other reviewers, first part)**
>
> # Ethics points raised by other reviewers
>
> Reviewers CS2n, DgCg, and Wvxh had no detailed ethics-related comments. Hence we will focus on the other three reviewers below.
>
> ## Reviewer pcKV
>
> > Overall, I find the level of ethical consideration in this paper sufficient but unsatisfactory. By clearly outlining the terms of use for the dataset, as well as employing state-of-the-art methods to filter and remove unsavory content from both the image and text portions of the dataset, this dataset maintains the current standard for ethical filtering. The method of using the CLIP embedding space to then train a model for NSFW detection, evaluated on a supervised dataset of only 1000 examples could be bolstered by comparisons with other NSFW detection techniques. Subsequently, a deeper analysis into the usage of certain vernacular that is more popular across different demographic groups, and how the filtering steps affect the prevalence of this content would prove extremely valuable.
>
> We thank the reviewer for suggesting further analyses. We agree that these are interesting directions and hope to pursue them in future work. We did not address them in our submission since adding multiple in-depth analyses would go beyond the scope of a single publication. For comparison, the ML community has by now published several analyses of the ImageNet dataset. Each of these analyses was a substantial research effort by itself and resulted in independent publications.
>
> To keep the amount of work for this paper manageable, which was already a large effort for a single publication, we therefore decided to focus on our main scientific goal: collecting a dataset for training large image-text models, and validating this dataset by actually training models and measuring their generalization capabilities. While doing so, we ensured that our submission meets current ethics standards in the field (as confirmed by the separate ethics review), but did not further develop these standards. While this is certainly a trade-off, we decided on this trade-off because our paper is primarily a paper on datasets, not on dataset ethics.
>
> Finally, regarding the NSFW detection we employed: any pointers to specific NSFW detection techniques would be helpful so we can extend our analyses to include the filtering techniques that would be most interesting to the reviewer. We hope that LAION can serve as a testbed and training set for future development of NSFW detection algorithms since the dataset contains a realistic snapshot of ML training data found on the web.
>
> ## Reviewer XLgX
>
> > The paper mentions very briefly that "after the release of LAION-400M, several groups investigated potential problems arising from an unfiltered dataset." These problems and criticisms merit more discussion than is given in the paper. It would be worth including a dedicated section in which you openly explain these points of contention and provide guidance about how to deal with issues of misogyny, pornography, and stereotypes in the data. It seems that this dataset paper has an obligation to discuss such issues openly, and this paper of the paper reads like something that is defensive rather than open to discussion.
>
> Thank you for this suggestion. We will extend our ethics section and related work to discuss these critiques of LAION in more detail.
>
> > Along these lines, it may be worth looking at multiple ways of tagging inappropriate content. Section 3.2 mentions that the two classifiers used for tagging are both CLIP-based; this may introduce some bias in the tagging/filtering that could be addressed with the use of other models or techniques. For example, CLIP may have difficulty detecting hateful memes or inappropriate text in images. Given the importance of this tagging to the behavior of downstream models, such issues merit thought and investigation.
>
> We share similar concerns and already tested our CLIP-based NSFW filtering approach with a validation set reviewed by human annotators. Section C.5 of the supplementary material contains additional details. We would also appreciate any suggestions for other models or techniques the reviewer has in mind so we can compare to them.
>
> > Additionally, the authors should probably set the default filter settings (e.g. for LAION-2B-en) in a very conservative manner, such that it has as little undesirable content as possible.
>
> The LAION-2B-en dataset contains fractional “unsafe” scores between 0 and 1 so that users of the dataset can control the trade-off between false positives and false negatives themselves (see https://huggingface.co/datasets/laion/laion2B-en-joined). For the web UI https://rom1504.github.io/clip-retrieval/ , we have indeed set a conservative default.

---

> ### Author Response · Authors · 2022-08-28
> **Ethics response, part 4 (responses to the ethics points raised by other reviewers, second part)**
>
> ## Reviewer XLgX, continued
>
> > It strikes me as insincere to say that users have full control over their data because they can remove their images from the internet.
>
> We appreciate the constructive criticism. We agree that removing oneself from the dataset will be above the technical level of most Internet users. We will update the datasheet accordingly and help users who contact us with specific removal requests to ensure that their URLs are removed from LAION. Please also refer to our response to the separate ethics review for more steps we are planning to mitigate the issue of identifiable humans in our dataset.
>
> > It would be good practice to describe how you intend to update the dataset when users revoke their consent to the use of their data.
>
> When users revoke their consent to the use of their data, we will publish an updated version of LAION without the respective URLs. This update may not happen right away but may instead be part of a monthly update to LAION.
>
>
>
> > A few of the answers on the dataset checklist should be more forthcoming. For example, there is a question "Q20: Does the dataset contain data that might be considered sensitive in any way" and the answer is misleading. The answer should be a straight "yes" -- the dataset is large enough that it will absolutely contain some sensitive data.
>
> We thank the reviewer for the specific and constructive feedback. We will update our response to Q20 and also make a pass over the rest of our dataset checklist to describe more potential harms.
>
>
> ## Reviewer hzjq
>
> > The authors have discussed this topic well in sections 3.2 and 7. Nevertheless, as the choice was to not remove from the dataset examples that could be "potentially offensive samples" in order to "encourage research in fields such as dataset curation" - which for me makes sense - and knowing the pairs came from Common Crawl dumps, I think that this work justifies further ethical discussion or review.
>
> We agree with the reviewer that leaving the potentially offensive samples in the dataset was not an easy decision, but reached the same conclusion as the reviewer. We are grateful for the separate ethics review and would welcome any further suggestions the reviewer has. We believe that publishing our entire data set generation process - including the filtering results and details - will lead to more transparency and higher quality datasets built from and around LAION-5B.

---

### Author Response · Authors · 2022-08-22
**Overall response**

We would like to thank all reviewers for the numerous in-depth comments. We really appreciate the combination of positive feedback and thoughtful, constructive suggestions for improving our paper and the LAION dataset.

Due to the volume of the reviews we received and the discussion timeline, we decided to release our responses to the technical points now. We are currently finishing our response to the more recent ethics review and will respond to the ethics review and ethics-related points raised by the reviewers by the end of Tuesday.

---

### Author Response · Authors · 2022-08-29
**Reminder: reviewer-author discussion ends on August 29th**

Dear Reviewers CS2n, pcKV, DgCg, and XLgX,

We'd like to reach out again to check if there were any additional questions or concerns about our rebuttal that we can address before the reviewer-author discussion period ends tomorrow (Monday) on August 29th.

Thanks again for taking the time to read our work and providing helpful feedback!

Paper Authors

---

### Meta-Review · Area_Chair_An13 · 2022-09-10

**Recommendation:** Accept
**Confidence:** 5

**Metareview:**

The reviewers agree that this dataset represents a huge undertaking that offers potential opportunities for various applications. The paper also provides insightful experiments across a series of use cases, models, and tasks. Several reviewers also noted the potential risks of the dataset and the potential for misuse -- although there is also an acknowledgment of these issues from the authors' side, and active steps to mitigate these risks. Authors are encouraged to incorporate the reviewer comments, especially some potentially factually incorrect statements raised by ethical reviewers surrounding IRB practices.

As it stands, this dataset should be of wide interest across various communities, including computer vision and natural language understanding. Given the careful considerations of the reviewers and the AC, this paper is recommended for acceptance in the program. The AC also recommends a more thorough review of similar resources before COCO and VisualGenome. There is the Pascal-1k, Flickr-8k, Flickr-30k, SBU-1M, IAPR-TC12 (ImageClef), which preceded most other works listed here.

---

### Decision · Program_Chairs · 2022-09-16

Accept